# Spontaneous assembly of redox-active iron-sulfur clusters at low concentrations of cysteine

Sean F. Jordan [1], Ioannis Ioannou[1,5], Hanadi Rammu [1,5], Aaron Halpern[1], Lara K. Bogart[2], Minkoo Ahn[3], Rafaela Vasiliadou[1], John Christodoulou [3], Amandine Maréchal [3,4] & Nick Lane [1✉]

Iron-sulfur (FeS) proteins are ancient and fundamental to life, being involved in electron transfer and $CO_2$ fixation. FeS clusters have structures similar to the unit-cell of FeS minerals such as greigite, found in hydrothermal systems linked with the origin of life. However, the prebiotic pathway from mineral surfaces to biological clusters is unknown. Here we show that FeS clusters form spontaneously through interactions of inorganic $Fe^{2+}/Fe^{3+}$ and $S^{2-}$ with micromolar concentrations of the amino acid cysteine in water at alkaline pH. Bicarbonate ions stabilize the clusters and even promote cluster formation alone at concentrations >10 mM, probably through salting-out effects. We demonstrate robust, concentration-dependent formation of [4Fe4S], [2Fe2S] and mononuclear iron clusters using UV-Vis spectroscopy, $^{57}$Fe-Mössbauer spectroscopy and $^1$H-NMR. Cyclic voltammetry shows that the clusters are redox-active. Our findings reveal that the structures responsible for biological electron transfer and $CO_2$ reduction could have formed spontaneously from monomers at the origin of life.

[1] Centre for Life's Origin and Evolution, Department of Genetics, Evolution and Environment, University College London, Darwin Building, Gower Street, London WC1E 6BT, UK. [2] UCL Healthcare Biomagnetics Laboratory, University College London, 21 Albemarle Street, London W1S 4BS, UK. [3] Institute of Structural and Molecular Biology, University College London, London WC1E 6BT, UK. [4] Institute of Structural and Molecular Biology, Birkbeck College, London WC1E 7HX, UK. [5] These authors contributed equally: Ioannis Ioannou, Hanadi Rammu. ✉email: nick.lane@ucl.ac.uk

ron–sulfur (FeS) proteins are fundamental to some of the most basic processes in life[1,2]. FeS proteins such as ferredoxin (Fd) enable $CO_2$ fixation in ancient autotrophic pathways, notably the acetyl CoA and reductive tricarboxylic acid (rTCA) pathways[3–6], as well as flavin-based electron bifurcation[7]. These pathways produce carboxylic acids, the universal core of intermediary metabolism in bacteria and archaea[8]. Phylogenetic[9], protein-folding[1] and comparative biochemical[6,10] analyses all suggest that ferredoxins are amongst the oldest proteins on Earth, predating the last universal common ancestor[1,9], and possibly even completion of the genetic code[11], although the conclusions drawn from phylogenetic work have been challenged[12]. Because the cofactors catalyse the redox chemistry, FeS clusters are likely to have been among the first catalytic structures enabling $CO_2$ fixation in early cells[2,10]. Yet despite the importance of FeS proteins, little is known about how the FeS clusters themselves arose in a prebiotic world.

The structure of canonical [4Fe4S] clusters is tantalisingly similar to the unit-cell of FeS minerals such as greigite (Fig. 1). Recent work shows that $CO_2$ reduction can be catalysed by FeS minerals, forming the same group of universally conserved carboxylic acids, including formate, acetate and pyruvate[13–15]. These prebiotic syntheses closely resemble microbial $CO_2$ reduction via the acetyl CoA pathway[13]. The overlapping catalytic properties of mineral surfaces and FeS proteins point to a possible transition from geochemistry to biochemistry at the emergence of life[2,10]. Yet the gap between inorganic FeS minerals and FeS proteins is still significant. On an atomic scale, FeS minerals are massive crystalline structures, whereas clusters are discrete entities, just a few atoms in size, which are therefore easily incorporated into proteins. If there was indeed a succession of steps in prebiotic environments from FeS minerals to FeS clusters, then how did these massive crystalline structures become 'downsized' to form biological FeS clusters? This gap would be much easier to cross if simple organic molecules such as monomeric amino acids could spontaneously interact with $Fe^{3+}$ and $S^{2-}$ to form FeS clusters capable of driving $CO_2$ fixation. Positive feedbacks could then drive and diversify organic synthesis, facilitating an 'autotrophic' origin of life[16,17]. Conversely, if FeS cluster formation requires complex polypeptides or even genetically encoded proteins then other processes would be needed to bridge the biosynthetic gap to a polymer world.

The formation of synthetic analogues of FeS clusters in the laboratory is a long-standing pursuit[18–22]. Most of these experiments involve complex syntheses performed in organic solvents such as dimethyl sulfoxide with little significance for the origin of life[18]. Less work has been done on FeS clusters from an origin-of-life perspective, although FeS clusters were recently synthesized through UV irradiation of the small peptide glutathione and other thiolate peptides[23]. These clusters required high concentrations of glutathione (240 mM) for their formation. Because the synthesis of high concentrations of peptides remains a challenging step in origin-of-life research, this work suggests that prebiotic syntheses of biological FeS clusters occurred relatively late and arguably not in any deep-sea hydrothermal setting. Another recent paper demonstrates the simultaneous synthesis of both thioesters and FeS clusters in water, yet the resulting clusters are likewise coordinated by peptide molecules[24]. Earlier work by Hill et al. did consider [4Fe4S] clusters formed by a 'capped' molecule of cysteine (N-acetyl-L-cysteine-N-methylamide), in which the charges on the amino and carboxylate groups are eliminated by acetylation and methylamidation, respectively, giving a non-charged coordination environment similar to peptides. But DePamphilis et al.[20] noted that N-acetyl-L-cysteine-N-methylamide has "proved difficult to obtain by direct synthesis" and so is unlikely to be prebiotically relevant. Evidence favouring the hypothesis that FeS clusters could spontaneously assemble from uncapped amino-acid monomers in anoxic systems containing FeS minerals such as greigite is still missing, and indeed is suggested to be unlikely on theoretical grounds[25].

Here, we show that cysteine—the predominant ligand for FeS clusters in modern proteins—can indeed form 'biological' FeS clusters in water under realistic prebiotic conditions. These FeS clusters form readily under anaerobic conditions at alkaline pH, with no need for UV light, and at surprisingly low concentrations of prebiotically relevant ligands. Mononuclear [1Fe0S], [2Fe2S] and [4Fe4S] clusters all form spontaneously from inorganic $Fe^{2+}/Fe^{3+}$ and $S^{2-}$ in the presence of L-cysteine alone (Cys-FeS). The clusters are stable in the absence of oxygen over 5 days. Bicarbonate ions seem to promote the coordination of purely inorganic FeS clusters and stabilize the assembly of Cys-FeS clusters under anoxic conditions. Cys-FeS clusters are redox-active, albeit their reduction potentials are higher than most modern ferredoxins. These findings reveal that the first step towards the synthesis of fundamental FeS proteins could occur

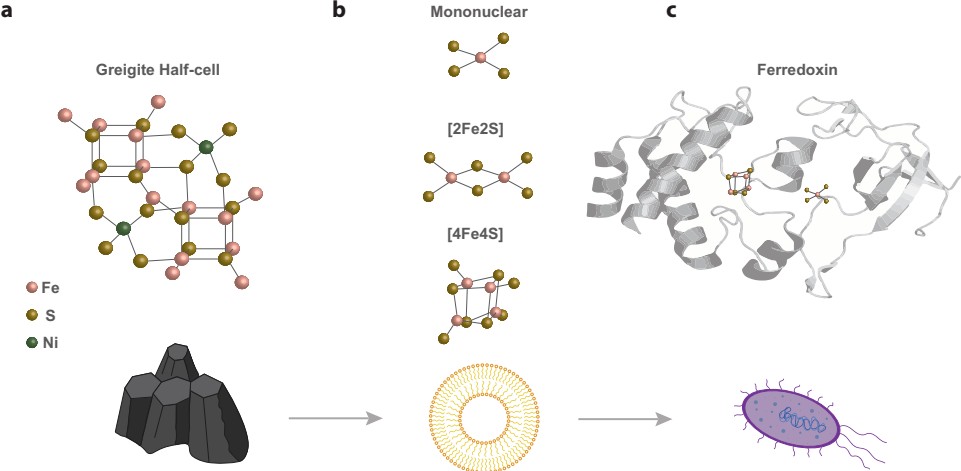

**Fig. 1 Similarity of FeS cluster structure in mineral, synthetic and biological forms. a** Half-cell structure of the FeS mineral greigite with [4Fe4S] units, **b** synthetic mononuclear [1Fe0S], [2Fe2S], and [4Fe4S] FeS clusters amenable to protocell incorporation, and **c** bacterial ferredoxin-thioredoxin reductase (PDB code: 1DJ7 [10.2210/pdb1DJ7/pdb]) structure containing mononuclear [1Fe0S] and [4Fe4S] clusters. For chemical structures: Fe atoms are pink, S are yellow and Ni are green.

spontaneously from low concentrations of simple monomers in water, giving a persuasive framework for an autotrophic origin of life.

## Results

**FeS clusters are formed from L-cysteine, FeCl$_3$ and Na$_2$S in water at alkaline pH**. We found that 5 mM L-cysteine, the predominant FeS cluster ligand in extant biology, could form FeS clusters under anaerobic conditions (<10 ppm O$_2$) in aqueous mixtures of 1 mM FeCl$_3$ and 1 mM Na$_2$S from pH 9 to 11, similar to what would be expected in Hadean alkaline hydrothermal vents[3,26–28]. The formation of FeS clusters was evident from the characteristic opaque-brown colour change and the UV-Vis spectra which displayed a prominent broad 420 nm signal, consistent with the formation of [4Fe4S] clusters[29].

We modelled the raw UV-Vis spectra following the method of Galambas et al.[30] to give indicative [4Fe4S] concentrations for each solution. This approach to quantitation is based on ligand-field theory, where the UV-Vis spectra are modelled using six Gaussian peaks, with <1.3% fitting error. We determined the integrated spectral intensities of the three lowest-energy ligand-to-metal charge

transfers (LMCT) for [4Fe4S]$^{2+}$, specifically peaks 3–5 (Fig. 2). Going from left to right, peak 3 (pink) is a high-energy LMCT band associated with thiolate S → Fe, while peaks 4 and 5 (green and blue) are intermediate energy LMCT bands associated with sulfides → Fe. Retaining the same 5:1:1 ratio of cysteine:Fe$^{3+}$:S$^{2-}$, we used reference spectra of known [4Fe4S]$^{2+}$ concentration to determine the concentration of Cys-FeS clusters in our samples, with cysteine concentrations ranging from 0.2 mM (Fig. 2a) up to 5 mM (Fig. 2b). We found a linear relationship between the concentration of cysteine and that of [4Fe4S]$^{2+}$ clusters over this range (Fig. 2c, black line). The spectral modelling calculations indicate that clusters formed at cysteine concentrations as low as 0.2 mM, with [4Fe4S]$^{2+}$ concentrations ranging from <10 μM up to >80 μM. The integrated curve areas and calculated uncertainty are shown in Fig. 2d, and the curve-fitted raw UV-VIS data for 0.4–4 mM cysteine are shown in Supplementary Figs. 1 to 7. Notably, the lowest concentration of cysteine analysed here (0.2 mM), formed Cys-FeS clusters at a concentration of 4.8 ± 0.1 μM when in a 5:1:1 ratio with Fe$^{3+}$ and S$^{2-}$ (i.e. 40 μM FeCl$_3$, 40 μM Na$_2$S). These low, micromolar concentrations clearly support the prebiotic plausibility of spontaneously formed Cys-FeS clusters.

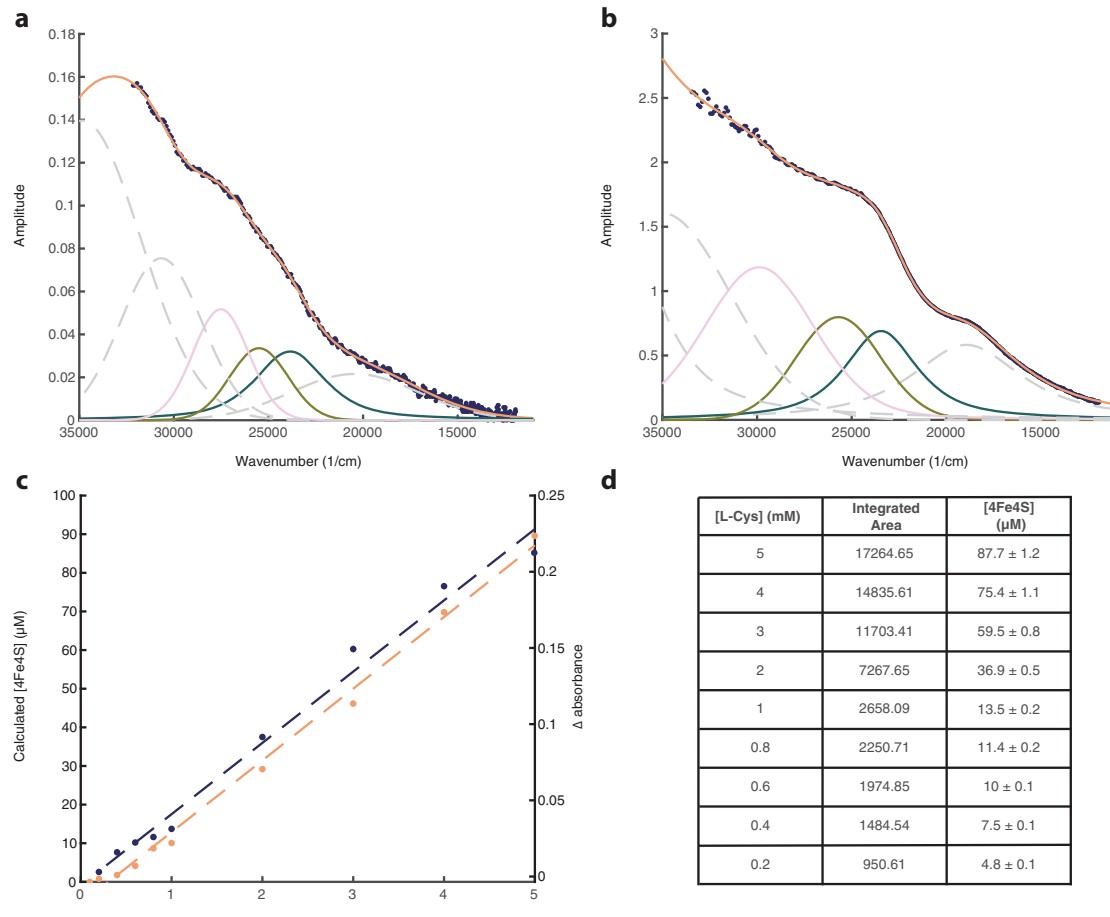

**Fig. 2 UV-Vis spectra of [4Fe4S]$^{2+}$ clusters, with curve fitting based on ligand-metal charge-transfer excitations. a** Curve-fitting based on the method of Galambas et al. (see 'Methods') recorded over the 35,000–10,000 cm$^{-1}$(285–1000 nm) spectral range for 0.2 mM L-Cys, with Fe$^{3+}$ and S$^{2-}$ in a 5:1:1 ratio. We fitted the three lowest-energy ligand-to-metal charge transfers (LMCT) for [4Fe4S]$^{2+}$: peak 3 (pink) is a high-energy LMCT band associated with thiolate S → Fe, while peaks 4 and 5 (green and blue) are intermediate energy LMCT bands associated with sulfides → Fe. **b**. Curve fitting for 5 mM L-Cys, with Fe$^{3+}$ and S$^{2-}$ in a 5:1:1 ratio. **c** Calculated concentration of [4Fe4S]$^{2+}$ clusters (μM) versus cysteine concentration (mM) based on integrated curve area (black), compared with a three-point method (brown, Δ absorbance; see text). **d** Table of data including calculated uncertainties based on reference [4Fe4S]$^{2+}$ concentrations from Galambas et al.[30]. For panels **a** and **b**, each peak corresponds to a specific excitation (from left to right: peak 1 grey dashed line—peptide charge-transfer; peak 2 grey dashed line—thiolate RS$^-$ ⟶ Fe; peak 3 pink line—thiolate RS$^-$ ⟶ Fe; peak 4 green line—S$^{2-}$ ⟶ Fe; peak 5 blue line—S$^{2-}$ ⟶ Fe; peak 6 grey dashed line—ligand field). Source data are provided as a Source data file.

For visual clarity, we focused our analysis on the change in the absorbance peak at 420 nm, allowing us to establish how a range of variables affect cluster formation (Fig. 3). We minimised the effect of the high background due to Rayleigh scattering by subtracting a linear baseline (black line) from the raw UV-Vis data (green line) (Fig. 3a). The baseline was anchored to two wavelengths (370 and 470 nm) so as to isolate the region of interest containing the main 420 nm signal for FeS clusters. The resulting spectra were then used to further characterise the samples based on this 420 nm signal (Fig. 3b). Using this method, a 5:1:1 ratio of L-cysteine:FeCl$_3$:Na$_2$S clearly formed Cys-FeS clusters at concentrations ranging from 5 to <1 mM cysteine (Fig. 3c, d). The validity of this method is corroborated by the linear relationship between the change in absorbance at 420 nm and cysteine concentration (Figs. 2c and 3d). A conservative minimum L-cysteine concentration for Cys-FeS cluster synthesis from this three-point analysis would be 0.8 mM, somewhat higher but in the same range as the fully-fitted model.

We then altered the concentrations of individual Cys-FeS components to test the extent to which they limited cluster formation. Varying L-cysteine concentration alone, while keeping FeCl$_3$ and Na$_2$S concentrations fixed at 1 mM, gave a lower limit of cluster formation around 3.5 mM L-cysteine, so an excess of L-cysteine is necessary for cluster formation (Fig. 4a). In fact, while 1 mM L-cysteine gave no FeS clusters under these conditions, the relationship with L-cysteine concentration was

ambiguous, with 3.5 mM L-cysteine producing greater absorbance than 10 mM, suggesting changes in the amount or diversity of FeS cluster species with differing ratios, which we confirm later. When keeping cysteine and Na$_2$S concentrations constant at 5 and 1 mM, respectively, FeCl$_3$ concentrations as low as 0.2 mM led to the formation of FeS clusters (Fig. 4b). In contrast, solutions with L-cysteine and FeCl$_3$ concentration held at 5 and 1 mM, respectively, but <0.4 mM Na$_2$S did not form clusters (Fig. 4c).

Ferric iron (FeCl$_3$) was found to be essential for FeS cluster formation in our experiments, as expected, and clusters did not form when ferric was substituted for ferrous iron (FeCl$_2$; Supplementary Fig. 8). Fe$^{2+}$ alone is less likely to form cubane structures such as greigite (corresponding in unit structure to [4Fe4S] clusters), tending to form layered minerals such as mackinawite instead (which correspond in structure to [2Fe2S] clusters)[31]. Our preliminary studies using optical density clearly showed that low-molecular-weight structures were being formed (not shown) but we did not pursue this further as the focus of this paper is on the synthesis of [4Fe4S] clusters. However, 1:1 mixtures of ferric and ferrous species did form FeS clusters coordinated by L-cysteine (Supplementary Fig. 9). Low concentrations of ferric iron are congruent with a Hadaean alkaline vent environment (see the 'Discussion' section). Importantly, regarding this potential setting, alkaline pH (>pH 9) was essential for Cys-FeS cluster formation. This is due to the p$K_a$ of the L-cysteine thiol

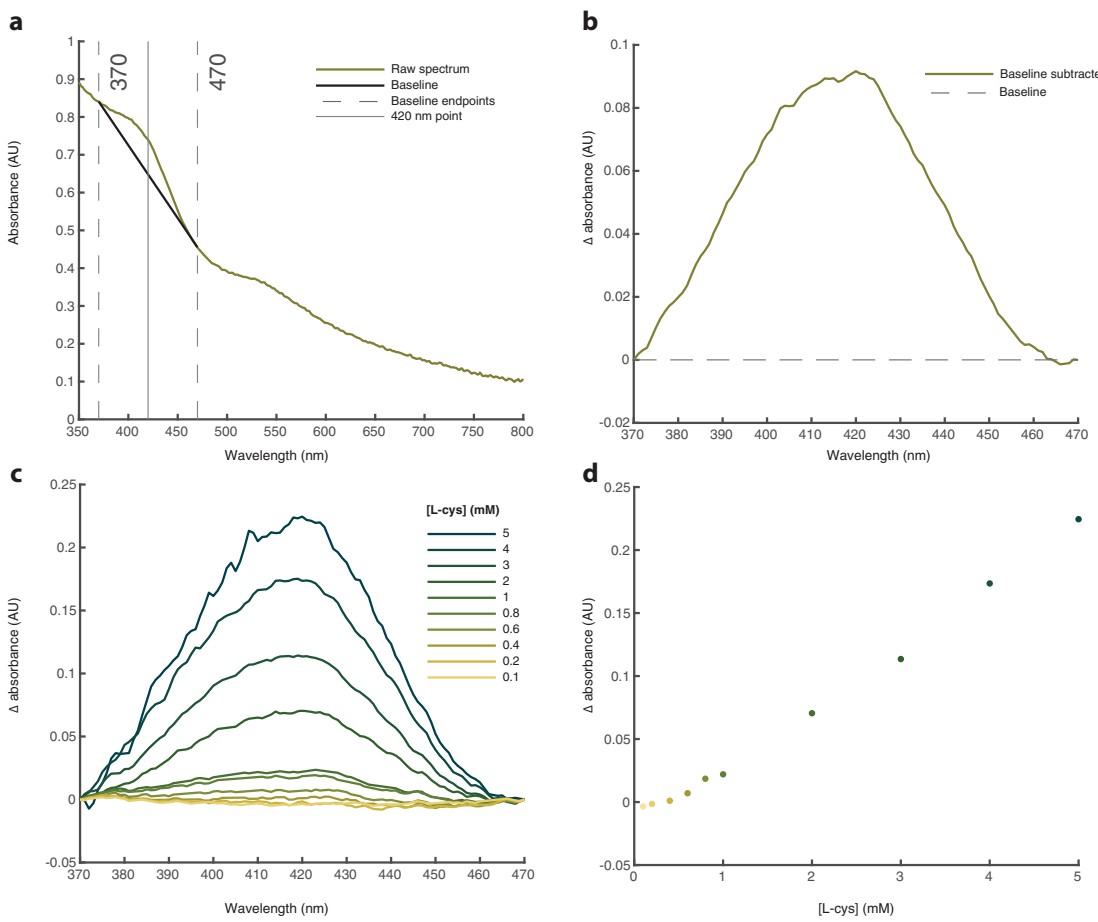

**Fig. 3 L-Cysteine promotes the formation of FeS clusters. a** Raw UV-Vis spectrum for solution containing 5 mM L-cysteine, 1 mM FeCl$_3$ and 1 mM Na$_2$S at pH 9. The parameters used to extract the 420 nm peak are also displayed. **b** the extracted 420 nm peak from the raw data in panel (**a**). **c** extracted 420 nm peak from spectra recorded from solutions containing increasing L-cysteine concentrations ranging from 0.1 to 5 mM at pH 9. Each solution was prepared using a 5:1:1 molar ratio of L-cysteine:FeCl$_3$:Na$_2$S. **d** Plot of the extracted 420 nm peak height as a function of L-cysteine concentration from data in panel (**c**). These values are compared with the full curve-fitting data used to establish concentrations in Fig. 2c. Source data are provided as a Source data file.

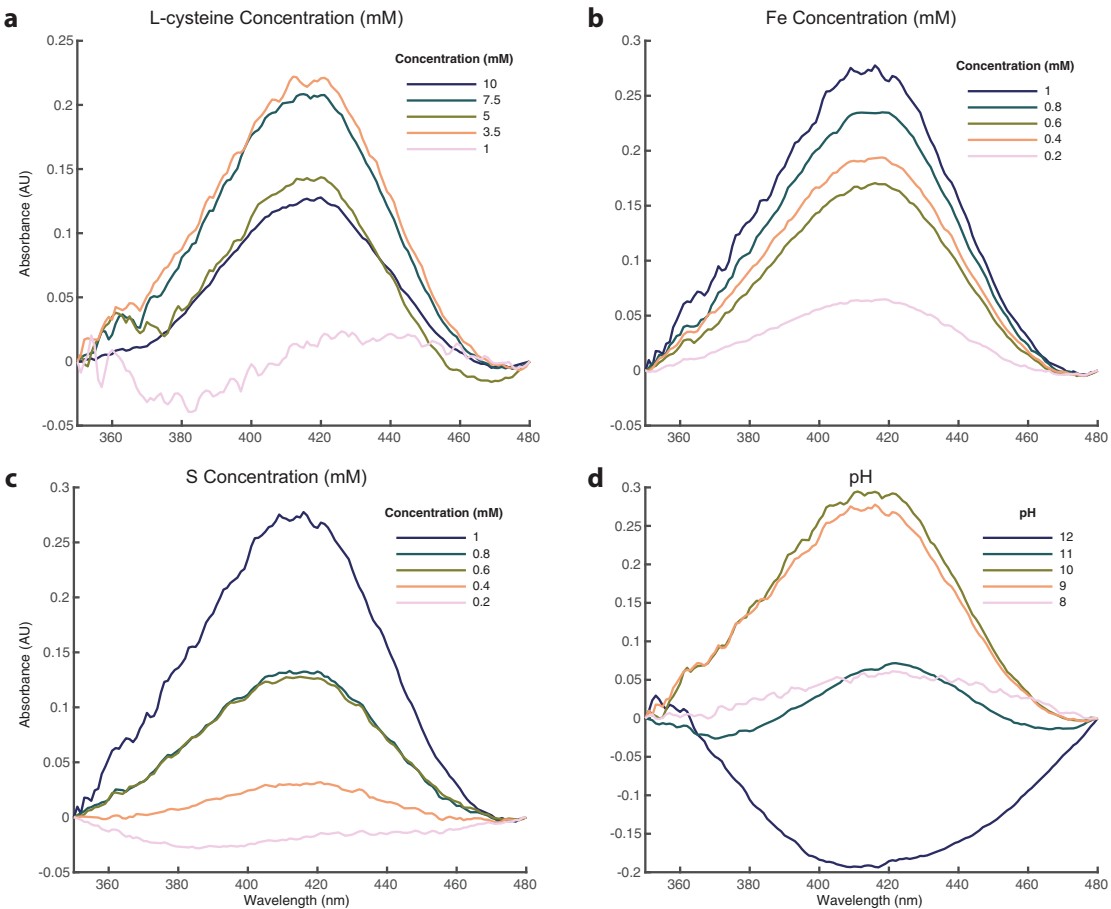

**Fig. 4 FeS clusters form at low concentrations of all components and across a wide pH range.** Extracted UV-Vis 420 nm peaks for pH 9 solutions prepared with **a** L-cysteine concentration ranging from 1 to 10 mM (concentrations of $FeCl_3$ and $Na_2S$ were both kept constant at 1 mM). **b** $FeCl_3$ concentrations ranging from 0.2 to 1 mM (L-cysteine and $Na_2S$ concentrations were constant at 5 and 1 mM, respectively). **c** $Na_2S$ concentrations ranging from 0.2 to 1 mM (L-cysteine and $FeCl_3$ concentrations were kept constant at 5 and 1 mM, respectively). **d** Extracted UV-Vis 420 nm peaks from solutions containing 5 mM L-cysteine, 1 mM $FeCl_3$ and 1 mM $Na_2S$ prepared at pH values ranging from 8 to 12. Source data are provided as a Source data file.

moiety, which is around 8.5 (ref. [32]). Because FeS clusters are coordinated by this functional group, the thiol must be deprotonated to form a bond with Fe ions. Accordingly, we observed Cys-FeS clusters between pH 9 and 11 (Fig. 4d). Cys-FeS clusters did not form at pH 12, which we attribute to the formation of Fe hydroxides being favourable above pH 11, as indicated by the formation of a white precipitate in these mixtures.

**FeS cluster species distribution is strongly affected by L-cysteine concentration.** As noted by Betinol et al.[33], [4Fe4S] clusters absorb strongly and tend to mask the presence of other clusters on UV-Vis spectroscopy. The concentration-dependence of [4Fe4S] cluster assembly on cysteine concentration and its tight correlation with the signature absorbance shoulder at 420 nm, suggests that other types of cluster were indeed masked by the presence of [4Fe4S] clusters. We therefore used room temperature [57]Fe Mössbauer spectroscopy to determine the proportions of individual species present[34]. The samples were prepared by lyophilization (see the 'Methods' section) as our initial preparations of $N_2$-dried samples did not produce normal UV-Vis traces when reconstituted (unlike our lyophilized samples). X-ray absorption spectroscopy has shown that the lyophilization procedure is fully reversible[35]. A combination of XANES and EXAFS spectra has shown a similar (and crucially, intact) iron coordination environment for the [4Fe4S] subsite of the

H-cluster in the freeze dried and in anoxic solution[35]. Likewise, Lorent et al. have demonstrated equivalent activity after lyophilization and reconstitution of $O_2$-tolerant [NiFe]-hydrogenases to freshly isolated enzyme, indicating that metal cofactors and amino acid side-chains responsible for proton/electron transfer were not altered by lyophilization'[36]. While most work to date on FeS proteins using Mössbauer spectroscopy has examined frozen biological samples[37–39], these considerations, coupled with earlier work on freeze-dried iron-oxide nanoparticles[40], and the findings reported below, suggest that lyophilization of prebiotic FeS clusters is an appropriate, if not yet fully validated, methodology.

L-cysteine concentration had a notable effect on both the range and relative proportion of FeS cluster species. Using 1 mM $FeCl_3$ and $Na_2S$ for all the Mössbauer experiments, we found that [4Fe4S] species only formed at the lower cysteine concentrations (3.5 and 5 mM) (Fig. 5). This double requirement for low cysteine concentration and alkaline pH presumably accounts for why earlier work did not detect FeS clusters when using much higher concentrations of cysteine at more neutral pH (7.3–8.1)[23].

Best fits to the data (black lines) were obtained using a number of Lorentzian shaped quadrupole doublets indicated by coloured lines (Fig. 5). The fitting parameters are provided in Table 1. Briefly, all spectra were least-squares fit to simultaneously fit all sub-spectra within a given sample, with all parameters left able to 'float'. The sub-spectra indicated with pink and blue lines dominate each panel of the figure. They have close isomer shifts

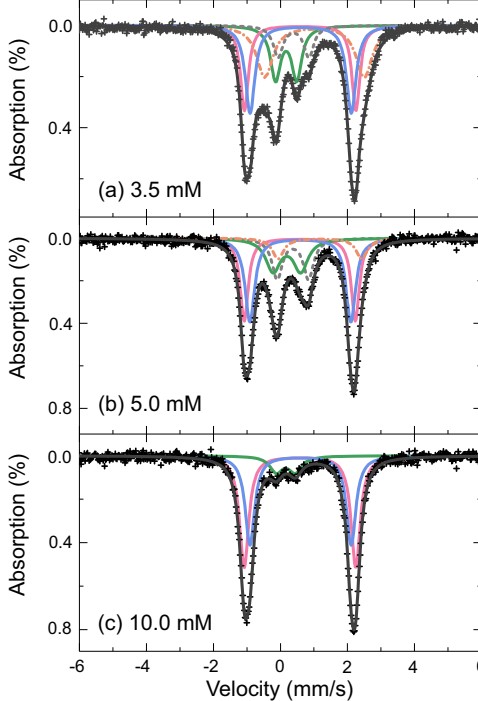

**Fig. 5 Mössbauer spectroscopy reveals a variety of FeS cluster species.** Room temperature $^{57}$Fe Mössbauer spectra of FeS clusters coordinated with **a** 3.5 mM, **b** 5.0 mM and **c** 10 mM cysteine ligand at pH 9 (FeCl$_3$ and Na$_2$S kept constant at 1 mM). All spectra were folded relative to αFe, with best fitting obtained using a least-squares fitting method, by adding a number of Lorentzian shaped doublets. Crosses represent observed data points with the black line showing best fit to the collected data. Each FeS cluster species is represented by a coloured line: mononuclear (pink line); [2Fe-2S]$^0$ (blue line); [2Fe-2S]$^{2+}$ (green line); [4Fe-4S]$^{2+}$ (grey dashed line); Fe$^{2+}$ with 5 or 6 coordination (orange dashed line). Mössbauer fitting parameters and related species are given in Table 1. Source data are provided as a Source data file.

of 0.58 and 0.60 mm/s, with high quadrupole splitting of 3.33 and 3.02 mm/s, respectively, which are characteristic of high-spin (S = 2) ferrous ions[41]. The isomer shifts are perhaps slightly low for Fe$^{2+}$, although this is probably due to the large electron delocalisation onto the sulfur ligands as observed for other Fe-Cys [2Fe-2S] bonded clusters[42]. The doublet with high quadrupole splitting (pink line) is indicative of valence localised Fe$^{2+}$ centres, most probably with quasi-tetrahedral sulfur coordination, i.e. mononuclear clusters[43], while the characteristics of the latter (blue line) are reminiscent of the high-spin cluster [2Fe-2S]$^0$, also known as the Rieske protein[37,44]. Our values agree well with mononuclear ferrous Fe in rubredoxin as reported by Schulz and Debrunner[45] (δ = 0.7 ± 0.02 mm·s$^{-1}$ at 4.2 K) and reduced rubredoxin in *Ch. Ethylica* reported by Rao et al.[38] (δ = 0.65 mm·s$^{-1}$ and ΔE$_Q$ = 3.16 mm·s$^{-1}$, both at 77 K) as well as the binuclear compounds reported by Leggate et al.[37] at 160 K; for example, tetrahedral ferrous sites in the [2Fe-2S]$^0$ cluster in *Aquifex aeolicus* ferredoxin δ = 0.71 mm·s$^{-1}$, ΔE$_Q$ = 2.75 mm·s$^{-1}$ (4.2 K)[46]. Second-order Doppler effects mean that δ increases as temperature decreases, and so temperature corrections must be performed in order to compare like with like. Finally, the very slight differences in the values of these two sub-spectra observed here between samples is likely a consequence of minor structural asymmetries around the two valence-delocalised Fe$^{2+}$ pairs, again also reported in the literature[43].

The sample prepared with 3.5 mM L-cysteine was best fit with a combination of five quadrupole doublets (Fig. 5a). Approximately 21% of the sample was comprised of mononuclear ferrous Fe clusters[23,39,43] (pink line) with an isomer shift (δ) of 0.59 mm/s (relative to αFe) and high quadrupole splitting (ΔE$_q$) of 3.33 mm/s. The sample also contained both [2Fe2S]$^0$ clusters[37,44] (26.4%, blue) and 17.2% [2Fe2S]$^{2+}$ clusters[23,43,44,47] (green). These components had isomer shifts of 0.6 and 0.2 mm/s and ΔE$_q$ of 3.02 and 0.58 mm/s, respectively, with the later species a likely product of oxidation of the former.

At 3.5 mM cysteine, ~9% of the overall sample composition comprised [4Fe4S]$^{2+}$ clusters (grey dashed line)[43,44,47], with an isomer shift of 0.35 mm/s and ΔE$_q$ of 1.01 mm/s, while the remaining 26.3% was attributed to ferrous iron with an expanded (either 5 or 6) coordination[44,48] (orange dashed line, δ of 1.02 mm/s and ΔE$_q$ of 2.97 mm/s). An increase in L-cysteine concentration to 5 mM doubled the proportion of [4Fe4S]$^{2+}$ clusters to ~18% (Fig. 5b). The concentration of mononuclear and both [2Fe2S] species remained largely stable, while the 5/6 coordinate species saw an appreciable reduction to ca. 9%. The most remarkable change in composition was observed when the L-cysteine concentration was increased to 10 mM (Fig. 5c). This resulted in a 50/50 mixture of mononuclear and [2Fe2S] species, with only ~10% of the latter represented by the oxidised clusters. There was no evidence of clusters with any higher co-ordinations with this higher concentration of L-cysteine, indicating that lower concentrations favour a higher diversity of FeS clusters.

**Bicarbonate forms purely inorganic FeS clusters and also stabilises Cys-FeS clusters.** In an effort to control the pH of the aqueous Cys-FeS cluster mixtures, sodium bicarbonate (NaHCO$_3$) was tested as a buffer. However, control samples with NaHCO$_3$, FeCl$_3$ and Na$_2$S in the absence of L-cysteine displayed the colour change and UV-Vis spectra indicating the presence of FeS clusters (Supplementary Fig. 10). We proceeded to investigate the possibility of inorganic cluster formation in the presence of bicarbonate by varying concentrations of NaHCO$_3$, FeCl$_3$ and Na$_2$S as well as pH. Bicarbonate ions clearly favoured the formation of [4Fe4S] clusters across a wide range of pH and concentration when FeCl$_3$ and Na$_2$S concentrations were below 0.5 mM. Specifically, bicarbonate promoted the formation of inorganic FeS clusters across a concentration range of 5–100 mM NaHCO$_3$, with an optimum concentration of 10 mM (Supplementary Fig. 10). Characteristic [4Fe4S] cluster UV-Vis spectra were observed across a pH range from pH ~8 to 11 (Supplementary Fig. 11). Attempts to characterise these samples further using $^{57}$Fe Mössbauer spectroscopy were hindered by the fragility of the samples during sample preparation. After lyophilisation, samples were brittle with a floss-like structure making it impossible to transfer to Mössbauer sample holders. We suggest that these inorganic clusters are formed due to the Hofmeister effect[49], whereby the Fe and S are essentially salted-out of solution. This interpretation is supported by similar, albeit weaker, effects with borate and phosphate buffers (Supplementary Fig. 12) but we did not pursue this further, as bicarbonate has far greater prebiotic relevance. These data highlight the need for caution when using buffers for prebiotic experiments.

To test the stability of Cys-FeS clusters in the presence of bicarbonate, samples were analysed by UV-Vis every 30 min for a total of 5 h. Even trace levels of oxygen in cuvettes sealed with parafilm (for measurements performed outside the anaerobic hood) led to the oxidation of Cys-FeS clusters, with the 420 nm signal disappearing over 3 h—indicating that the Cys-FeS clusters had broken down. Samples containing 10 mM bicarbonate

**Table 1 Best fit $^{57}$Fe Mössbauer parameters obtained from the 295 ± 5 K spectra shown in Fig. 4.**

| Sample | δ (mm/s) | ΔE$_q$ (mm/s) | w3 (mm/s) | S (%) | Ion species | Coordination |
|---|---|---|---|---|---|---|
| FeS 3.5 mM L-cysteine | 0.20 | 0.58 | 0.18 | 17.2 | Fe$^{3+}$ | [2Fe-2S]$^{2+}$ |
| | 0.35 | 1.01 | 0.17 | 9.0 | Fe$^{3+}$ | [4Fe-4S]$^{2+}$ |
| | 0.59 | 3.33 | 0.14 | 21.1 | Fe$^{2+}$ | Mononuclear, 4 co-ord |
| | 0.60 | 3.02$^a$ | 0.17 | 26.4 | Fe$^{2+}$ | [2Fe-2S]$^0$ |
| | 1.02 | 2.97 | 0.29 | 26.3 | Fe$^{2+}$ | 5 or 6 co-ord |
| FeS 5 mM L-cysteine | 0.25 | 0.67 | 0.23 | 15.1 | Fe$^{3+}$ | [2Fe-2S]$^{2+}$ |
| | 0.37 | 0.93 | 0.19 | 18.3 | Fe$^{3+}$ | [4Fe-4S]$^{2+}$ |
| | 0.58 | 3.32 | 0.15 | 25.7 | Fe$^{2+}$ | Mononuclear, 4 co-ord |
| | 0.60 | 3.01 | 0.17 | 31.4 | Fe$^{2+}$ | [2Fe-2S]$^0$ |
| | 1.08 | 2.70 | 0.23 | 9.4 | Fe$^{2+}$ | 5 or 6 co-ord |
| FeS 10 mM L-cysteine | 0.16 | 0.57 | 0.23 | 9.8 | Fe$^{3+}$ | [2Fe-2S]$^{2+}$ |
| | 0.58 | 3.33 | 0.17 | 49.5 | Fe$^{2+}$ | Mononuclear, 4 co-ord |
| | 0.60 | 3.02 | 0.17 | 40.7 | Fe$^{2+}$ | [2Fe-2S]$^0$ |

The best fits were obtained using Lorentzian shaped doublets, and fit using a least-squares method in the Recoil fitting programme.
$^a$Indicates fixed parameter during fitting.

remained stable throughout the entire 5-h period, compared to 3 h for those without (Fig. 6). Clearly, the presence of bicarbonate imparts a protective effect to the Cys-FeS clusters, delaying their oxidation and breakdown.

This stabilisation was confirmed by NMR spectroscopy (Fig. 7). The paramagnetic relaxation enhancement (PRE) effect of Fe$^{3+}$ results in the absence of a signal for free L-cysteine in solution (Fig. 7a). However, when FeS clusters are formed with L-cysteine, the PRE effect is reduced allowing for the detection of both Hα and Hβ protons from the bound L-cysteine as observed with increasing L-cysteine concentrations (Fig. 7b). Increasing bicarbonate concentration (10–50 mM) while maintaining 5 mM L-cysteine, increased the Hα and Hβ signals attributed to bound L-cysteine, suggesting that bicarbonate promotes further incorporation of L-cysteine into FeS cluster complexes (Fig. 7c). The saline conditions expected in an oceanic hydrothermal vent environment may therefore have assisted in the assembly of stable FeS clusters at the origin of life. Monitoring of the Cys-FeS clusters under anoxic conditions showed that the clusters were stable for up to 14 h with increasing bound L-cysteine over time (Fig. 7d). This not only suggests that the Cys-FeS clusters are stable for long periods of time, but that they also increase in number. The same analysis performed under oxic conditions showed a decrease in bound L-cysteine over time, with almost undetectable signal after 17 h (Fig. 7e).

The discrepancy in temporal stability of FeS clusters as measured by UV-Vis versus NMR analysis was probably due to the different sample holders. Cuvettes for UV-Vis were sealed with parafilm stretched across the opening: this stretching is likely to increase the porosity of the parafilm, allowing more O$_2$ to enter the cuvette. In contrast, the NMR tubes were capped with lids designed for anaerobic analyses which could potentially maintain the anoxic headspace above the sample indefinitely. Moreover, the headspace above the sample within the NMR tube is of a larger volume than the sample itself: this large anoxic headspace likely delayed O$_2$ penetration on exposure to aerobic conditions when compared to the relatively small headspace of the UV-Vis cuvettes. For these reasons, the NMR results are most reflective of the actual temporal stability of the Cys-FeS clusters which not only survive but also continue to form under anoxic conditions for upwards of 14 h. Indeed, when maintained in the anaerobic hood, Cys-FeS clusters remained stable for up to 5 days when investigated by UV-Vis spectroscopy as well (Supplementary Fig. 13).

**Cys-FeS clusters are electrochemically similar to higher potential Fd proteins.** Cyclic voltammetry (CV) was performed on Cys-FeS samples with L-cysteine concentrations of 3.5, 5 and 10 mM

while maintaining both FeCl$_3$ and Na$_2$S at 1 mM. All three samples produced a reduction potential peak in the range expected for Fd protein, albeit closer to the high-potential iron–sulfur protein (HiPIP) range[50]. Cyclic voltammetry measurements were made using a Ag/AgCl (3 M NaCl electrolyte) reference electrode (+209 mV vs. SHE). At 3.5 and 5 mM cysteine, Cys-FeS clusters both produced reduction peaks at −400 mV while at 10 mM cysteine, the Cys-FeS clusters were slightly less reducing, at −310 mV (Fig. 8). Both 3.5 and 5 mM solutions provide voltammograms with ΔEp values of 55 and 63 mV, respectively, consistent with a reversible reaction and a Nernst $n$ value of 1 (refs. [19,51]). The reduction potential values display a linear shift with increasing scan rate (Supplementary Figs. 14–19) and the ratio of anodic peak current to cathodic (Ia/Ic) is close to 1 (3.5 mM = 1.06; 5 mM = 1.49). This is indicative of a quasi-reversible reaction. In contrast, the 10 mM Cys-FeS voltammogram is closer to an irreversible reaction with a ΔEp of 78 mV and a much larger anodic current (47.9 μA) compared to cathodic (−11.6 μA) (Ia/Ic = 4.1). The unusual wave shape of the anodic signal in all three voltammograms likely reflects a combination of the complexity of these solutions, each one containing multiple FeS species as evidenced through Mössbauer spectroscopy, and possibly some adsorption to the electrode during analysis (Supplementary Figs. 6–11). Cathodic signals gave more reproducible reduction potential values for each solution, although they provide limited diagnostic information as the peaks are a composite of the multiple species present. The lower reduction potentials recorded with clusters formed from 3.5 to 5 mM cysteine (Fig. 8) corresponded to the highest proportions of [4Fe4S]$^{2+}$ clusters (9–18% from Mössbauer spectroscopy), whereas 10 mM cysteine did not contain any [4Fe4S]$^{2+}$ clusters at all (Fig. 8 and Table 1). This suggests that [4Fe4S]$^{2+}$ clusters (the most ubiquitous in biology[31]) are more highly reducing than [2Fe2S] or mononuclear Fe clusters when chelated by cysteine alone. This tentative conclusion is supported by the fact that the dominant species at all concentrations of cysteine tested here was the [2Fe2S]$^0$ cluster (Table 1). As noted earlier in relation to Mössbauer spectroscopy, this cluster resembles the higher potential Rieske proteins, with reduction potentials of −150 mV or higher relative to the SHE[52]. However, the redox potentials of individual species, whether chelated by cysteine alone or by short peptides such as glutathione, requires systematic further study, which we will address in future work.

## Discussion

Our results show that FeS clusters of biological significance can form spontaneously at alkaline pH (pH 9 to 11) from μM concentrations of Fe, S and the single amino acid L-cysteine (Figs. 2–4). We have characterised their structure through UV-

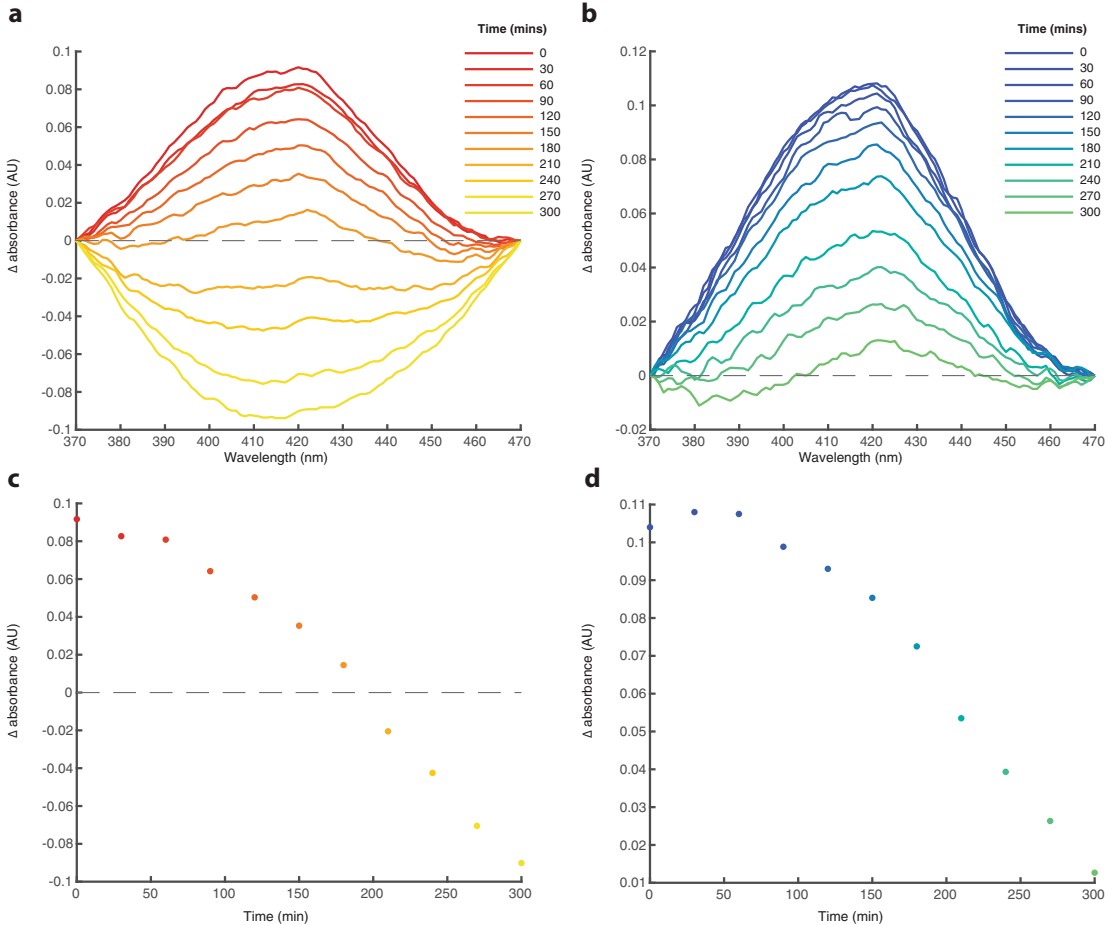

**Fig. 6 Bicarbonate stabilises Cys-FeS clusters in the presence of $O_2$. a** Extracted UV-Vis 420 nm peaks for Cys-FeS cluster solutions prepared with 5 mM L-cysteine, 1 mM $FeCl_3$ and 1 mM $Na_2S$ analysed every 30 minutes for 5 h. **b** as in panel **a** with 10 mM $NaHCO_3$ included in solutions. **c** Plot of the extracted 420 nm peak heights as a function of time from data in panel (**a**). **d** Plot of the extracted 420 nm peak heights as a function of time from data in panel (**b**). Source data are provided as a Source data file.

Vis and Mössbauer spectroscopy, and show that a mixture of mononuclear [1Fe0S], [2Fe2S], [4Fe4S], and higher coordinate geometry clusters form under these conditions (Fig. 5). Strikingly, [4Fe4S] only form at lower cysteine concentrations (down to a mM ratio of about 3.5:1:1 cysteine:$Fe^{3+}$:$S^{2-}$) whereas higher concentrations or ratios (such as 10:1:1 cysteine:$Fe^{3+}$:$S^{2-}$) do not form [4Fe4S] clusters. These Cys-FeS clusters are equivalent to those found in extant proteins, notably ferredoxins, which are ancient and were most likely present in the earliest cells[1,2,6,9,10,53]. Cyclic voltammetry shows that the Cys-FeS clusters are redox-active, albeit their reduction potentials are higher than most modern ferredoxins, and closer to the range of high-potential iron–sulfur proteins[30]. The cathodic signals reflect a composite mixture of several different types of FeS cluster. Those containing more [4Fe4S] clusters gave lower redox potentials (Fig. 8), also as observed in Fd proteins[30]. Monitoring Cys-FeS cluster stability by $^1H$ NMR spectroscopy demonstrated stability over at least 14 h (Fig. 7), while UV-Vis spectroscopy suggests that [4Fe4S] clusters are stable for 5 days under strictly anoxic conditions (Supplementary Fig. 13). These Cys-FeS clusters are stabilised further by the presence of bicarbonate, which can even form inorganic [4Fe4S] clusters in the absence of any organic ligand (Fig. 6). Thus, our results suggest that FeS clusters could have formed readily under Hadean submarine alkaline hydrothermal systems, potentially driving $CO_2$ reduction and protometabolism at the origin of life.

Hadean alkaline hydrothermal vents probably incorporated FeS minerals such as greigite into their chimney structures, given the chemical composition of anoxic oceans and serpentinization fluids on the early Earth[26,53]. The cubane unit-cell structure of greigite is affine to that of biological [4Fe4S] clusters, and has been shown to facilitate $CO_2$ reduction under prebiotic conditions, forming carboxylic acids including formate and acetate[13]. Use of a low overpotential on greigite additionally generated methanol and pyruvate[15] (which in Preiner et al. were also synthesised using a magnetite catalyst[13]). Theoretical thermodynamics indicates that Hadean $CO_2$ reduction should favour the production of carboxylic acids, fatty acids and amino acids[54]. Other Krebs-cycle intermediates and amino acids have indeed been formed under these conditions, albeit using iron as an electron donor[8]. Previous experimental work shows that hydrothermal Fischer-Tropsch-type syntheses can also form long-chain 1-alkanols and fatty acids[55], which spontaneously assemble into protocells under alkaline hydrothermal conditions[56,57]. Amino acid synthesis from pyruvate has also been demonstrated under similar conditions[58]. However, the polymerisation of amino acids and nucleotides in water remains elusive, suggesting that the earliest stages in the emergence of life might have taken place in a monomer world[17]. Thus, protocells formed from combinations of monomeric molecules such as fatty acids were arguably the earliest units of selection at the origin of life[16,59,60]. While FeS clusters have previously been formed under prebiotic conditions

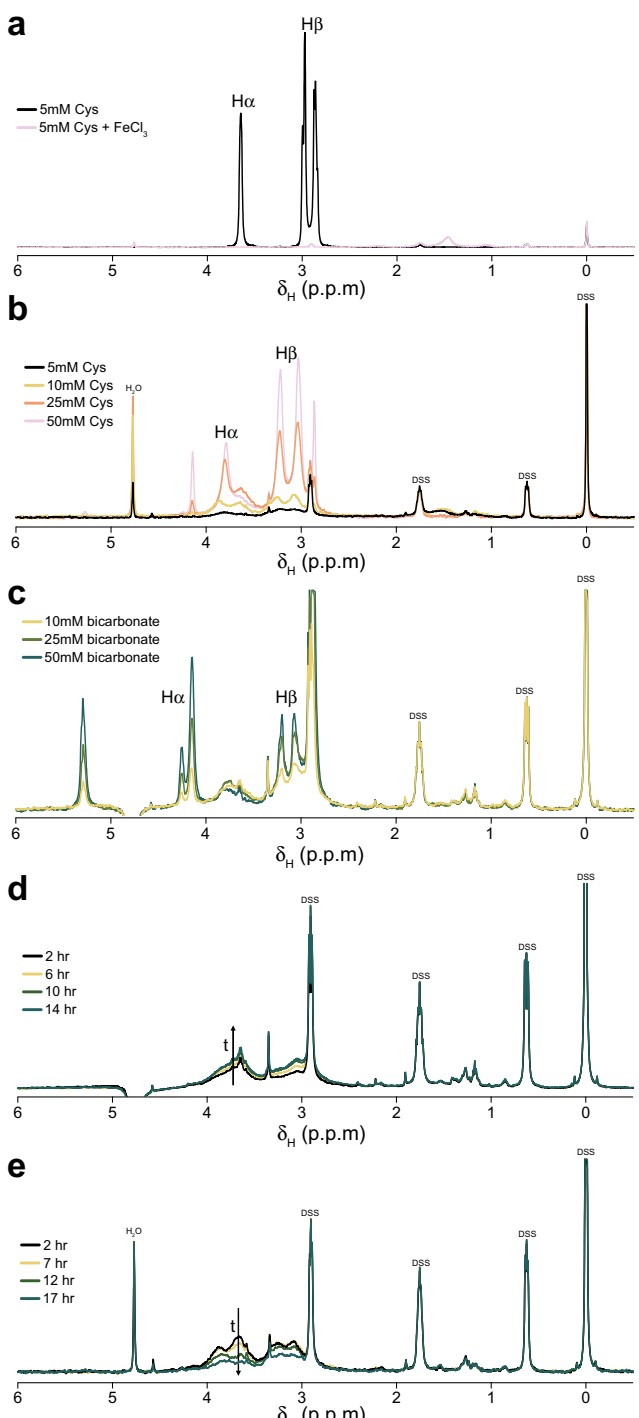

**Fig. 7 Cys-FeS clusters continue to form over time under anoxic conditions.** $^1$H NMR spectra showing **a** the effect of Fe on free cysteine in solution essentially removing the cysteine signal. **b** Cys-FeS clusters prepared with increasing cysteine concentrations leading to increased signal for Hα and Hβ of bound cysteine. **c** HCO$_3$⁻-FeS clusters prepared with increasing NaHCO$_3$ concentrations leading to increased signal for Hα and Hβ of bound cysteine. **d** 5 mM Cys-FeS clusters in anoxic conditions over 14 h showing an increase in cysteine incorporation, and **e** 5 mM Cys-FeS clusters in the presence of oxygen over 14 h showing a decrease in bound cysteine. All solutions were prepared at pH 9. Source data are provided as a Source data file.

from high concentrations (240 mM) of glutathione[23], such high concentrations of specific peptides are problematic in most origins of life scenarios, at least at an early stage—we of course anticipate a later stage in which [4Fe4S] clusters are chelated by short non-coded polypeptides. The key question is therefore whether biological FeS clusters could form from low concentrations of monomers that could associate with protocells.

A monomer world operating in alkaline hydrothermal systems predicts that single amino acids such as cysteine could form FeS clusters. L-cysteine is the predominant FeS cluster ligand in proteins across all extant life[61]. It is likely that this relationship has been conserved since its origin. Cysteine is unique among all amino acids as it is the only thiol-bearing proteinogenic amino acid[62]. This factor makes cysteine ideal for FeS cluster formation, particularly at alkaline pH. With a p$K_a$ of ~8.5, the cysteine thiol is deprotonated at higher pH values[32], allowing it to coordinate FeS clusters. Nonetheless, theoretical investigations of FeS-cluster maquette chemistry suggested that [4Fe4S] cluster formation depends on the secondary structure of short peptides, and molecular dynamic simulations imply that monomeric cysteine molecules could not form [4Fe4S] nests[25]. Our Mössbauer results show that some [4Fe4S] clusters do indeed form, as well as mononuclear [1Fe0S], [2Fe2S], and some 5 and 6 coordinate FeS species in solutions containing L-cysteine. Full curve-fitting of UV-Vis spectroscopy data suggests that [4Fe4S] clusters could form at cysteine concentrations as low as 200 µM, with a minimum of 40 µM FeCl$_3$ and Na$_2$S at pH 9 (Fig. 2). The concentration of cysteine used here is two to three orders of magnitude lower than the concentration of the small peptide glutathione used in earlier prebiotically relevant work[23,63].

The reduction potential of our Cys-FeS clusters varied from −300 to −400 mV relative to the Ag/AgCl electrode (depending on cysteine concentration) or about −100 to −200 mV relative to the SHE. These are cathodic values. The midpoint reduction potentials were harder to estimate given the non-reversibility of the mononuclear clusters that predominated with 10 mM cysteine, or the quasi-reversibility at lower concentrations of cysteine (Fig. 8; Supplementary Figs. 14–18). The most clearly reversible redox cycle was at 3.5 mM cysteine, where the cathodic and anodic peaks nearly mirrored each other at high scan rates (0.8 V/s; Supplementary Fig. 14), giving a midpoint reduction potential of about −250 mV relative to the Ag/AgCl electrode (about −50 mV relative to the SHE). At face value this is clearly not low enough to reduce CO$_2$. However, as noted, the cyclic voltammograms reflect the contribution of multiple FeS species, which are likely to vary in behaviour (reversibility) and reduction potential. One of the two dominant FeS species detected at all concentrations of cysteine by Mössbauer spectroscopy was the [2Fe2S]$^0$ cluster. This is reminiscent of the Rieske protein clusters that are characterised by higher reduction potentials, in the range of −150 to +400 mV relative to the SHE, and might partly account for the relatively high redox potentials measured here. In particular, we note that the lower cysteine concentrations (3.5 and 5 mM) produced 9–18% [4Fe4S] clusters, whereas higher cysteine concentrations (10 mM) gave no [4Fe4S] clusters at all (Fig. 5). The cathodic peak on cyclic voltammetry was slightly broader at the lower concentrations of cysteine, potentially reflecting a range of reduction potentials in a composite peak of multiple species (Fig. 8a vs c). If so, then the increased contribution of [4Fe4S] clusters might have lowered the cathodic reduction potential from around −310 to −400 mV (vs Ag/AgCl electrode; Fig. 8). More systematic work involving purification and maturation of FeS clusters will be needed to resolve the redox potentials of these individual species, especially the [4Fe4S] clusters measured here, both with cysteine alone and with short peptide

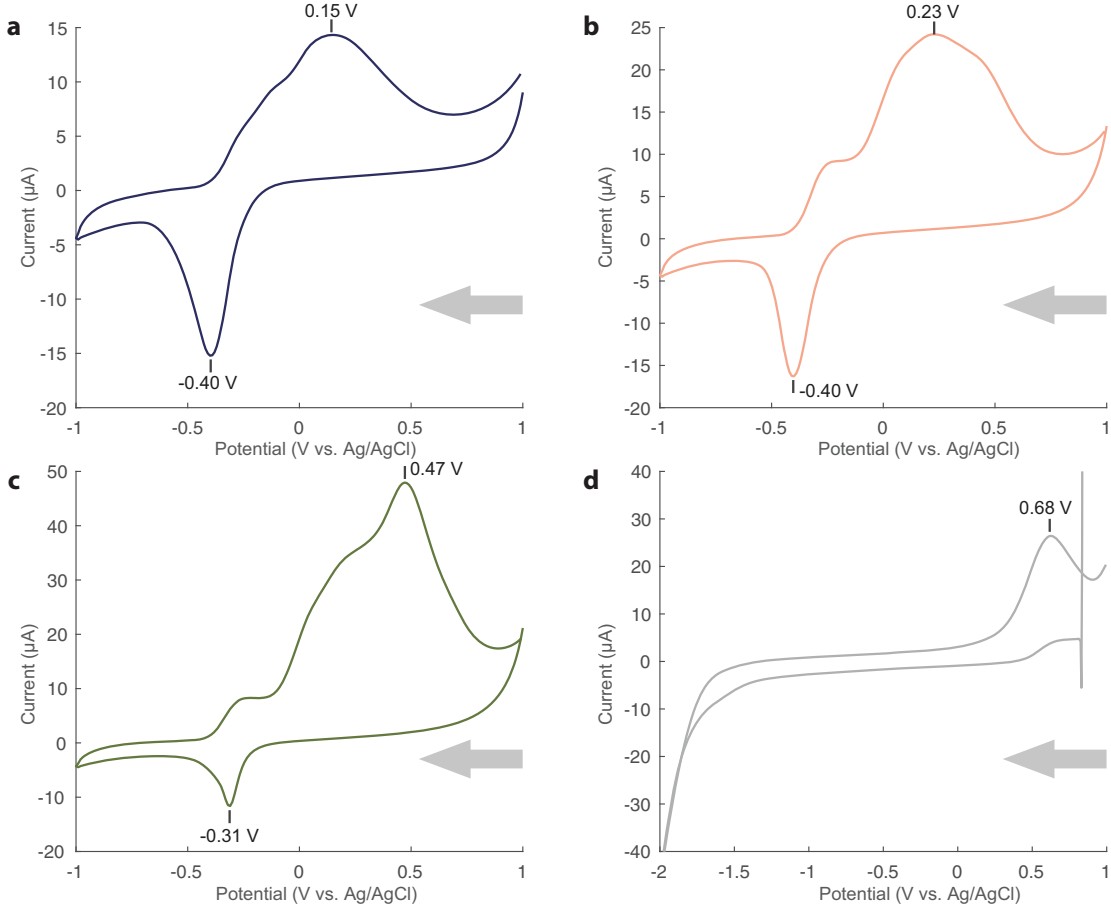

**Fig. 8 Reduction potentials of Cys-FeS clusters reflect those of Fd proteins.** Cyclic voltammograms of Cys-FeS clusters formed at pH 9 in the presence of **a** 3.5 mM, **b** 5 mM or **c** 10 mM L-cysteine, with FeCl₃ and Na₂S at equimolar 1 mM concentrations. **d** 10 mM L-cysteine alone. Cyclic voltammograms were performed at a scan rate of 100 mV/S. Potentials (V, versus Ag/AgCl (3 M NaCl)) are marked for the apex of each cathodic and anodic peak. Arrows indicate the scanning direction. Source data are provided as a Source data file.

nests, and this is planned for a separate paper. We anticipate that some of these species could have a redox potential low enough to reduce CO₂.

In principle, therefore, the same chemistry that promotes greigite formation under hydrothermal conditions could also form FeS clusters, which could mediate CO₂ reduction, hydrogenation and carbonylation reactions in protocells, driving growth. Such incorporation of FeS clusters into protocells paves the way for protometabolism linked to positive feedback loops, favouring simple membrane heredity through growth and division[16]. In this view, polymerization is more likely to occur at higher concentrations of monomers, especially if associated with protocell surfaces[64,65]. The availability of cysteine is a moot point. Only a few studies have investigated prebiotic cysteine synthesis. Hennet et al. used a variety of mineral catalysts likely present in Hadean vent environments (e.g. pyrite, magnetite, etc.) and produced trace amounts of cysteine under those conditions[66]. Parker et al. outlined two possible mechanisms for prebiotic synthesis of cysteine from glycine[67]. A key requirement for both of these pathways is H₂S, which can be formed through the serpentinization reactions that feed alkaline hydrothermal vents[2,53,68]. It has also been shown that cysteine could theoretically be synthesised from mercaptoacetaldehyde via the Strecker reaction[62]. A plausible precursor for mercaptoacetaldehyde is glycoaldehyde, a key intermediate of the formose reaction that occurs readily under simulated hydrothermal vent conditions[69]. More recently, Foden et al. reported a prebiotic synthesis of

cysteine in water with high-yield[70]. Therefore the availability of cysteine at the origin of life is becoming increasingly plausible. In a carboxylic-acid rich monomer world in alkaline hydrothermal vent systems, cysteine should be the favoured FeS cluster ligand, just as it is in biology. We used L-cysteine rather than a racemic mixture for analytical comparative purposes only; we see the emergence of chirality as a separate question that we have not addressed here.

The availability of ferric iron in Hadean hydrothermal systems is also uncertain. While we could not form clusters from ferrous iron alone, we did detect clusters produced by a 1:1 mix of ferrous and ferric iron (Supplementary Fig. 9). Assuming that the Hadean oceans were ferruginous[71], ferrous iron could be oxidised to nanoparticulate ferric iron by photochemical reactions or oxidants such as NO derived from volcanic emissions, meteorite impacts or lightning strikes[72]. Recent work suggests that Hadean oceans were much less ferruginous than had been thought, and argues early banded-iron formations were formed instead by local precipitation and in situ oxidation of hydrothermally derived ferrous iron[73]. Nonetheless, even in this case, hydrothermal systems would still be rich in ferrous, and potentially ferric, iron. Thermodynamic modelling indicates that alkaline hydrothermal conditions can partially oxidise ferrous to ferric iron at temperatures above about 70 °C, implying that the simple mixing of alkaline hydrothermal fluids with seawater within such vents could promote the continuous cycling between ferrous and ferric iron[74]. If so, then we would anticipate that hydrous ferric

chlorides would be available at the low μM concentrations required to form [4Fe4S] clusters.

Perhaps more surprisingly, our results suggest possible FeS cluster formation when Fe and S are in solution with $HCO_3^-$ alone (Fig. 6). This serendipitous discovery occurred when we used $HCO^{3-}$ as a buffer to maintain the pH of Cys-FeS solutions. The control sample that did not contain L-cysteine also displayed a smaller 420 nm peak when analysed by UV-Vis. This anomalous result was repeated multiple times and it was confirmed that the characteristic spectrum for [4Fe4S] clusters was indeed produced by solutions of 10 mM $HCO^{3-}$, 500 μM $FeCl_3$ and 500 μM $Na_2S$ (Fig. 6). Attempts to characterise these clusters by Mössbauer spectroscopy were unsuccessful as the sample preparation by lyophilisation resulted in a mixture that was too brittle to work with. EPR spectroscopy should allow for further characterisation of these solutions in future. We suspect that the clusters are formed due to the Hofmeister effect, where the Fe and S are essentially 'salted-out' of solution[49], leading to increased interactions between the species that favour cluster coordination. This inference is supported by the fact that other buffers, including phosphate and borate, also produced shoulders at 420 nm, suggestive of [4Fe4S] cluster formation (Supplementary Fig. 12). Inorganic aqueous FeS clusters similar to these have been observed in multiple modern environments[75–78]. In fact, they are believed to account for a major fraction of dissolved metal species in anoxic marine and freshwater environments including deepsea hydrothermal vent systems[79].

In addition to forming clusters independently, we observed that $HCO_3^-$ stabilised Cys-FeS clusters. Cys-FeS clusters in disposable cuvettes sealed with parafilm were stable for 3 h, as indicated by the deterioration of the 420 nm signal during UV-Vis analysis (Fig. 6). Cys-FeS clusters that formed in the presence of 10 mM $HCO^{3-}$ were stable for an additional 2 h under identical conditions (Fig. 6). Likewise, $^1H$ NMR analysis showed an increase in bound L-cysteine peaks in solutions containing additional $HCO^{3-}$ (Fig. 7). It is possible that the Hofmeister effect[49] due to the presence of $HCO_3^-$ leads to increased incorporation of L-cysteine into FeS clusters. This interpretation is supported by the greater UV-Vis absorbance values observed for Cys-FeS clusters with $HCO_3^-$ in solution (Fig. 6). Thus, $HCO_3^-$ not only stabilises FeS clusters but also promotes their formation. The combination of inorganic $HCO_3^-$-FeS and organic Cys-FeS clusters shown here to form readily in simple aqueous solutions suggests that mineral clusters would have been abundant in marine environments on the early Earth, making their incorporation into newly forming protocells even more likely.

The association of FeS clusters with membranes could enable protocells to reduce $CO_2$ directly, driving 'autotrophic' growth. Modern hydrogenotrophic methanogens utilise membrane-bound Fe(Ni)S proteins such as the Energy-converting hydrogenase (Ech) to reduce ferredoxin, which in turn reduces $CO_2$ via the acetyl CoA pathway to form acetyl CoA (the basis for fatty acid and isoprene synthesis required for membranes) and much intermediary metabolism[6,57,80–82]. The reduction of ferredoxin depends on the proton-motive force, with ion gradients generated through electron bifurcation coupled to the methyltransferase complex (Mtr)[83]. The reduction potential of two [4Fe4S] clusters in Ech is pH-dependent, facilitating electron transfer from $H_2$ to ferredoxin[84,85]. In principle, geochemically sustained proton gradients could drive $CO_2$ reduction through analogous steps[86]. Recent work from Hudson et al.[14] shows that pH gradients across inorganic FeS barriers do indeed facilitate $CO_2$ reduction, as does pressure, which both point to a submarine location. Previous work shows that synthetic [4Fe4S] clusters are capable of $CO_2$ reduction to hydrocarbons[87], although to our knowledge this has not as yet been accomplished under prebiotic conditions. We are actively working on these questions: how do simple FeS clusters interact with prebiotic membranes, how does their redox potential vary with pH gradients, and can they catalyse $CO_2$ reduction under prebiotic conditions?

In conclusion, FeS clusters are essential for electron transfer and $CO_2$ reduction in biology. They are unquestionably ancient biological cofactors, and their simplicity and fundamental roles fit them as central to the emergence of life. Before the origin of genes and proteins, simple molecular systems such as monomeric Cys-FeS clusters could have driven protometabolism. We have shown here that stable FeS clusters form spontaneously in solutions of L-cysteine, $FeCl_3$ and $Na_2S$ at the low concentrations expected in prebiotic submarine alkaline hydrothermal systems. These monomeric Cys-FeS clusters are redox-active and have structures exactly equivalent to those found in modern ferredoxins, albeit more work needs to be done to determine the reduction potentials of individual species. Our findings demonstrate that the first steps from inorganic mineral catalysts to some of the most fundamental biological cofactors are surprisingly easy under widespread prebiotic conditions.

## Methods

**Formation of clusters**. All synthesis experiments were performed in triplicate at room temperature, under anoxic conditions in a Coy Lab type B vinyl anaerobic chamber with an atmosphere of 5:95% $H_2:N_2$ gas mix and Pt catalysts for $O_2$ removal. Deoxygenated water was used as the starting solvent for all solutions and was prepared by purging 1 L of Milli-Q water with anhydrous $N_2$ for 1 h.

For FeS cluster synthesis an aqueous solution containing L-cysteine and $Na_2S$ was prepared with deoxygenated deionized $H_2O$ in a beaker with a magnetic stirrer and adjusted to pH 9 with a 2 M NaOH solution. Ferric chloride hexahydrate ($FeCl_3 \cdot 6H_2O$) was added and the solution was brought to a final volume with deoxygenated deionised $H_2O$ to provide required concentrations of all reactants. After the addition of $FeCl_3$ the colour of the reaction mixture turned quickly to brown, indicating the formation of FeS clusters. Bicarbonate FeS clusters were prepared following the same method with L-cysteine replaced by $NaHCO_3$. For assessing the stabilising effect of bicarbonate on Cys-FeS clusters, $NaHCO_3$ was added prior to L-cysteine in the above procedure.

**Ultraviolet-visible (UV-Vis) spectroscopy**. UV-Vis absorption spectra of freshly prepared solution mixtures were acquired using a Thermofisher NanoDrop 2000c and recorded using NanoDrop PC software (v1.4.0.1). A sample (1.5 mL) of each reaction mixture was transferred to a plastic semi-micro cuvette inside the anaerobic chamber. These were then sealed with parafilm to minimize oxygen penetration during transport between the hood and nanodrop. All samples were recorded in triplicate immediately after preparation. The broad peak at 420 nm, where [4Fe4S] clusters absorb, was extracted from the raw spectra by subtraction of a linear baseline between 370 and 470 nm. Data processing was carried out using Microsoft Excel and MATLAB (version R2020b).

The modelling of spectral features is based on Galambas et al.[30]. Briefly, these authors fitted three reference spectra[22,88,89] using a combination of six peaks to describe the spectral features; Fe d → d ligand-field (1 broad peak), $S^{2-}$ → Fe (2 peaks), thiolate $RS^-$ → Fe (2 peaks), and peptide (1 peak) charge-transfer excitations in the 12,500–37,500 $cm^{-1}$ regions. Because the peaks relating to peptide coordination were not relevant to our analysis, we instead determined $[4Fe4S]^{2+}$ concentration by comparing the integrated spectral intensities of peaks 3–5 with those of the three reference spectra of known concentration, hence the associated uncertainties. Peak fitting was carried out using the PeakFit software programme (version 4.12, Seasolve Software, Framingham, MA, USA) with no background correction, no smoothing, and Gaussian/Lorentzian (G/L) combination line shapes without any constraints to amplitude, position, line-width and G/L mixing.

**Nuclear magnetic resonance (NMR) spectroscopy**. NMR samples (600 μL final volume) were prepared in 5 mm NMR tubes (Sigma-Aldrich) by dissolving FeS samples in the respective buffer supplemented with 10% $D_2O$ (v/v) and 0.001% DSS (w/v) as an internal chemical shift reference. Experiments were recorded at 298.2K. Proton (1H) spectra were recorded on a Bruker Avance II 600 MHz spectrometer equipped with a TXO cryogenic probe or a Bruker Avance III 500 MHz spectrometer equipped with a TCI cryogenic probe. All spectra were processed using Topspin v3.5 software (Bruker).

**$^{57}$Fe Mössbauer spectroscopy**. $^{57}$Fe Mössbauer spectroscopy measurements were performed at room temperature using a SeeCo W302 spectrometer (SeeCo Inc., USA) that was operated in the constant acceleration mode. Spectra were recorded

using the Mössbauer Spectral Analysis Software from SeeCo. (WMOSS4F). One hundred microlitres of sample was lyophilized (to preserve the integrity of samples without causing any defects or sample artefacts) ground with boron nitride and sealed in a 2.1 cm diameter Mössbauer sample holder with epoxy glue. All preparation was performed inside the anaerobic chamber. Samples were mounted in transmission geometry, with a $^{57}Co$ in Rh foil as the source of the 14.4 keV γ-rays. Velocity calibration was performed by recording a reference spectrum from a 10-μm-thick foil of αFe, also at room temperature.

All spectra were folded and baseline corrected using cubic spline parameters derived from fitting the αFe calibration spectrum, following a protocol implemented in the Recoil analysis programme[90]. All spectra were fit with multiple sub-spectra using Lorentzian lineshapes. For all samples, all parameters were allowed to float unless otherwise indicated in Table 1. Mössbauer data was analysed using Recoil Mössbauer Spectral Analysis Software[90].

**Cyclic voltammetry (CV)**. Experiments were conducted under ambient conditions, using an EmStat (PalmSens), single-channel potentiostat. Cluster solution mixtures were prepared inside the anaerobic chamber. All solutions were purged with $N_2$ for 15 min prior to analysis, to ensure that no $O_2$ was trapped during the transfer from the anaerobic chamber to potentiostat. A conventional three-electrode cell composed of a glassy carbon working electrode, a Ag/AgCl reference electrode (Alvatek, RE-5B 3 M NaCl, +209 mV vs. standard hydrogen electrode (SHE)) and a platinum wire counter electrode was used for the electrochemical measurements. We did not explicitly add a supporting electrolyte, to minimize interference with cluster formation (as occurred with bicarbonate). However, ~0.01 M NaCl was present in samples through acid-base titrations. Relatively low ionic strength electrolytes, notably NaCl, have been shown to diminish coagulation of FeS nanoparticles[91], and NaCl is clearly of prebiotic relevance in marine environments. The working electrode was cleaned with 0.05 μm aluminium oxide slurry on a polishing cloth and thoroughly rinsed with double-distilled $H_2O$ before each experiment. Data were recorded using PSTrace software (v5, PalmSens).

**Statistics and reproducibility**. All UV-Vis and CV analyses were performed in triplicate and individual spectra/traces represent the average of three measurements. Modelling results are representative of analysis of triplicate datasets. Mössbauer spectra are representative of >500,000 scans recorded over >120 h acquisition. For NMR samples, a high number of transients (≥256) were recorded and accumulated for each spectrum to ensure high signal-to-noise ratio (>10) for all peaks that were analysed.

**Reporting summary**. Further information on research design is available in the Nature Research Reporting Summary linked to this article.

## Data availability

All data generated in this study are provided in the Article, Supplementary Information and Source data files. Source data are provided with this paper.

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

## Acknowledgements
Where possible, the scientifically-derived colour scheme Batlow[92] was used for the preparation of figures. We are grateful to the BBSRC (N.L., BB/V003542/1 and H.R., LIDo Doctoral Training Programme) and bgc3 for funding. A.M. is funded by the Medical Research Council U.K. (Career Development Award MR/M00936X/1 and Transition Support MR/T032154/1).

## Author contributions
S.F.J., I.I. and N.L. conceived the experimental approach; N.L. and S.F.J. supervised the project; S.F.J., I.I., H.R. and A.H. did the UV-Vis spectroscopy; H.R. and A.M. did the full curve UV-Vis analyses; L.B., I.I. and S.F.J. did the Mossbauer spectroscopy; M.A., J.C., I.I. and S.F.J. did the NMR spectroscopy; R.V. and I.I. did the cyclic voltammetry; S.F.J., H.R. and N.L. prepared the figures; N.L. and S.F.J. prepared the first draft of the paper; all authors contributed to the writeup.

## Competing interests
The authors declare no competing interests.
