## [Peer Review File · Nature Communications]

Spontaneous assembly of redox-active iron-sulfur clusters at low concentrations of cysteineEditorial Note: Parts of this Peer Review File have been redacted as indicated to remove third party material where no permission to publish were obtained

REVIEWER COMMENTS

Reviewer #1 (Remarks to the Author):

The paper by Jordan et al addresses the abiotic assembly iron-sulfur (FeS) clusters in conditions compatible to scenarios of the origin of life at alkaline hydrothermal vents. Starting with L-cysteine and FeCl₃(Fe III) - pH between 9 and 11 - 4Fe-4S, 2Fe-2S and Fe are formed. Moreover the authors characterise the FeS centers and show that their redox potential is similar to the ones found in extant ferredoxins. Moreover, FeS were also found to be assembled in the absence of Cysteine, in mixtures containing HCO₃, Fe(III) and Na₂S.

This is a very exciting paper showing the formation of a cofactor essential to all life forms in prebiotic conditions and I am impatiently waiting for the future studies regarding the EPR characterization of the FeS clusters.

I only have a few comments that I would like the authors to address, all related with the Fe(III) availability in alkaline hydrothermal systems.

In the condition tested, and in the absence of additional information, Fe(III) from FeCl₃ is in the soluble state.

What was the initial redox potential of the mixture (or before adding S-compounds) that allowed the non-precipitation of iron?

Also, please extend the discussion regarding the presence and/or evidences (if available), for the formation of iron-III minerals or the presence of Fe-III in solution (harder I guess) under prebiotic
Reviewer #2 (Remarks to the Author):

In the paper by Jordan et al., the formation of iron-sulphur clusters using the amino acid cysteine is investigated. The main question addressed is the concentration dependences (including the concentrations of Cysteine, Fe, HS⁻, and pH) required to form these clusters. The techniques used to probe this are uv/vis spectroscopy, Mössbauer spectroscopy, NMR. Cyclic voltametry was used to investigate the reduction potential of the species formed.

From a purely chemical standpoint, the work conducted is important because it sheds light on what a free amino acid might be doing in the cell (though metal ion concentrations there are very low), what a free amino acid might be doing in the environment, and as was the focus of the authors - what free amino acids might be doing before the origin of life, en route to potentially facilitating the origin of life. FeS clusters in biology function in within proteins in catalytic, structural, and redox activities. These are very broad functions, and it seems difficult to imagine biology as we know it without FeS clusters. It is a key question in the origin of life to understand redox transformations, because fundamentally today, biology is an electronic property, with all organisms moving electrons between molecules to do work (i.e synthesize the polymers that are needed for cell doubling).

Acknowledging the above, significant work has been done towards understanding the abiotic formation of FeS clusters, perhaps most recently experimentally by the Mansy group. One key weakness of this former work - as the authors duly note - is the high concentrations of thiol used in the experiments which is often around 100mM. It seems quite high for "prebiotic" chemistry... Herein lies the value of the authors current approach; by systematically investigating the concentration requirements for FeS cluster formation, the authors are paving new ground and supplying relevant chemical information for the origin of life, when the authors supply evidence that ~800µM Cys is needed to have a uv/vis observable cluster.

Despite being enthusiastic about the provenance of this work, I have major concerns about the conclusions drawn and the data acquisition performed to reach these conclusions which should be significant enough to prevent publication of the work. I hope my comments below will allow the authors to dive deeper into their question and gain the insight they are working towards.

Although the concentration dependences of FeS cluster formation with Cysteine are a valuable part of the paper, the main conclusion of the article is that cysteine affords "monomeric Cys-FeS clusters have structures and reduction potentials exactly equivalent to those found in modern ferredoxins.". As I outline below, insufficient data are presented to justify this claim. (it is a quibble, but worthwhile to note that it is not clear what "monomeric" means here, since 4Cys are needed for 1 cluster and clusters made of both less and more than 4Fe would be considered to be monomers from a structural perspective)

Major concerns:

Cysteine: There is a strong theoretical basis against this conclusion. As an example, in a series of papers, the Holm group used the acetylated, methyl amine carboxy substituted version of cysteine (acetyl-cysteine-N-methylamide) to investigate FeS cluster formation from free amino acids, for example:

Hill et al, Journal of the American Chemical Society / 99:8 / April 13, 1977

In these works, they specifically used this "capped" version of the amino acid, because free amino acids with their negatively charged carboxyl group and their positively charged amino group are expected to behave fundamentally different from an amino acid that is incorporated into a peptide. Following the work by the Holm group has detailed the critical role that intrinsic/extrinsic solvent effects can have on FeS cluster redox, it is difficult to accept that free Cys would behave like the peptide, or like the "capped" version of Cysteine referenced above. While I have not been able to find the relevant redox properties and cluster formation when free Cys was used, but it is likely published (note that the paper referenced above is part 15 in a series). The authors need to provide a much more detailed background of the literature to convince this reviewer and other readers that Cysteine is a faithful mimic of cysteine containing peptides.

uv/vis spectroscopy:

The authors rely on uv/vis spectroscopy to prove the formation of FeS clusters in the presence of cysteine. This is fair since it is quite easy to use and provides useful information. It is however very unfortunate that only a limited data range is supplied for only a portion of the spectrum - from 370nm-470nm. This is a severe limitation, because single iron "rubredoxin" like complexes usually appear around 330nm (e.g. DOI: 10.1039/c3dt53157k) and 2Fe complexes often show absorbances at longer wavelengths (e.g. Ugulava et al Biochemistry, Vol. 40, No. 28, 2001). Indeed, there are a host of absorbance features that can be observed for different cluster types (Hagen et al. J. Am. Chem. Soc. 1983, 105, 3905-3913)

With this in mind, it seems dramatically insufficient to report only 100nm of the spectrum.

Mössbauer spectroscopy:

Mössbauer spectroscopy is a very powerful and appropriate tool for studying iron, and it has been used extensively in the study of FeS proteins. As noted in the Pandelia et al review paper cited by the authors "Typical requirements for biological samples of Fe-containing proteins are volumes of 250 - 400 μ L and ^{57}Fe concentrations of 1–5 mM [12]".

In the current paper however, the sample was analyzed in a dried state (which the authors call lyophilized, but I suspect is actually simply dried?).

Analyzing a dry sample is fundamentally different from analyzing a wet sample. Specifically, there is no reason at all to suspect that the iron coordination in water would represent anything at all of what is occurring in the solution phase. In the analysis of FeS clusters by groups who use Mössbauer, it is to

this reviewer's knowledge always done in the wet phase, and this reviewer has never seen a paper where a dried state was used, because it is widely accepted that dry samples are not useful for understanding liquid. Therefore, the analysis of dry samples is inappropriate and cannot be used to analyze the coordination, or redox state of the aqueous clusters.

It might be worthwhile to compare the dried material with Cys to material without Cys, but at any rate, information about Cys-Fe in the aqueous phase is not accessible when dry. It is also improbable that the full reduced 2Fe cluster (blue line) exists in the experiments, where at least partially oxidized (+1, +2) species might be expected.

Quickly checking the 2Fe₂S clusters in the Pandelia paper, I find that the velocities associated with the absorbances are not at the values shown in the paper for the 2Fe species. Going further, both $|\Delta E_Q|$ (mm/s) and δ (mm/s) values are different from what Pandelia et al reported in their table 1 in comparison to table 1 in the current paper. This is a major inconsistency and makes the data and interpretation suspect, even in the absence of the major problems associated with analyzing and interpreting coordination in dry samples. Perhaps the numbers simply do not match because they are reporting different things.

cyclic voltametry:

Cyclic voltametry is a powerful tool for understanding the tendency of FeS clusters (and any number of other species) to accept and give electrons. The Hill paper cited above used the technique to good effect, for example. In the current work, the authors took samples from the glove box and "rapidly transferred to the potentiostat". This seems rather unfortunate, since it is almost certain that oxygen is now a major factor in the observations.

Putting aside the concern of O₂ however are the interpretations. The authors say that "edge" features exist which show that the clusters have potentials comparable to modern ferredoxins (~600mV vs AgCl written by the authors), but I cannot find the "edge" features, and cannot conclude with the authors on this statement. Instead, I suspect that the authors are seeing ferrous/ferric transitions, together with multiple redox active sulfur steps, especially in the more oxidized regions of the cathodic scans. It's very worthwhile here to visit the FeS literature, for example: Bura-Nakić et al.

Electroanalysis 2011, 23, No. 6, 1376 – 1382

minor comments:

with regards to citations 1 and 9 cited for claiming that ferredoxins are ancient. It is worthwhile to note that (1) conflated spatial proximity within proteins for antiquity - an interesting yet unproven hypothesis and that (9) seems to have been proven largely incorrect (doi:10.1093/molbev/msaa089). The authors might be better off simply claiming that it's hard to imagine biology without these clusters, as I have done in the beginning of this review.

"For any form of selection to operate, there had to be a continuous succession of steps leading from Fe²⁺/Fe³⁺ and S²⁻ mixing in solution (which spontaneously crystallize to form FeS minerals) to early cells, where discrete FeS clusters were assembled within proteins instead."

This sentence is very unclear to me, can the authors work on it to make it more specific and concrete?

"Moreover, the headspace above the sample within the NMR tube is of a larger volume than the sample itself, so it would have taken significantly longer for O₂ to diffuse into the sample than in the UV-Vis cuvette." This is another unclear sentence, since volume might not be related to the rate of oxidation...

"Both 3.5 and 5 mM solutions provide voltamograms with ΔE_p values of 55 and 63 mV respectively, which correspond to a one-electron transfer"

No, this suggests that the charge transfer has an n value of 1 in the Nernst equation. It means that the number of protons equal the number of electrons:

When $Ox + ne^- + mH^+ \rightarrow Red$
the slope at 25 °C will be: $-(m/n) \times 59 \text{ mV/pH}$.

consider the case of no protons being involved in the reaction; the slope would be zero.

“orders of magnitude lower than that of the polymers used in earlier work¹⁷”

It is correct that Mansy's group uses high concentrations, and as I pointed out above, a major strength of this current work is that it sets out to explore the concentration limit for FeS cluster formation, which to the knowledge of the current reviewer has not been done.

I quickly checked work from Cowan's group "[dx.doi.org/10.1021/ja302186j](https://doi.org/10.1021/ja302186j)" which seems to be a forerunner of Mansy's work and found 10mM. But in a recent paper that discussed FeS clusters I also found that FeS clusters seemed to have been made at "750 mM Ferredoxin peptide", which is equal to the concentration of cysteine used in the current work (though it is very important to remember that the ferredoxin forms multiple linkages with the cluster from a single molecule, and cysteine probably is not formally a ligand but instead requires 4 per cluster (if excluding the possibility of NH_3^+ and carboxyl ligation, which could be important).

It is worthwhile to ask if anyone has done a concentration dependence test for peptides as well...

Methods:

problems with oxygen:

in the uv/vis: sealing with parafilm is not sufficient to minimize oxygen reaction. It means that all of the results obtained with uv/vis have a strong oxygen component, which is in turn important for which clusters would be formed. Measurements in cells with screw caps, or better yet, within the anaerobic chamber, would be preferable.

in the potentiostat: it again seems that the measurements are going to be dominated by the presence of O_2 .

Figure 1: This reviewer can't think of a ferredoxin that has both a 4Fe and a single Fe unit. Can the authors please indicate the pdb used in generating this?

Reviewer #3 (Remarks to the Author):

Iron-sulfur clusters are considered as central cofactors in biochemistry, and also ones whose evolutionary history is likely to date back to the very emergence of life from prebiotic chemistry. It is unknown, however, how Fe-S clusters may have been produced under prebiotic conditions in a way robust and simple enough to have enabled their facile incorporation into the earliest forms of nascent biochemistry. The most prominent example to date is the one reported in Nat Chem by Bonfio et al, and involves UV light-mediated photooxidation of Fe^{2+} driving the synthesis of [2Fe-2S] and [4Fe-4S] clusters, stabilised by cysteine-containing tripeptides such as glutathione.

In the present paper by Jordan et al, the synthesis of [2Fe-2S], [4Fe-4S], as well as mononuclear clusters is shown under much simpler conditions (no UV light needed), at low Fe^{2+} and Cys concentrations and at a mildly alkaline pH – which is all in line with the hydrothermal vent hypothesis explored by the authors of the manuscript. The iron-sulfur clusters are characterised by Mössbauer spectroscopy, UV-Vis spectroscopy and 1H NMR, and their redox properties are examined using cyclic voltammetry.

The analytical part of the study appears very clear and thorough, and the results well-documented and easy to follow. Only for this reason, i.e. the demonstration that iron-sulfur clusters can be formed using such remarkably simple chemistry, I am of the opinion that this paper deserves publication. Without the need to invoke the UV light (therefore a process that could only have taken part in very shallow waters) or the presence of complex thiols like glutathione, this study indicates that the

simplest forms of ferredoxin-type “protocofactors” may have predated the emergence of stable oligopeptides or robust lipid membranes – and why not already taken part in catalysis? Here, I need to stress that this is not to say the two approaches are mutually exclusive, even if one is likely to have preceded the other, and I think it would be worthwhile to make it more explicit in the text.

Before publication, I suggest that the authors address the following questions and remarks that should hopefully improve the manuscript. I will list them below in a chronological order:

- Page 3 – the following paper by de Leeuw and co-workers could be added to the discussion on the use of greigite in CO₂ reduction and fixation: Roldan, A. et al. Bio-inspired CO₂ conversion by iron sulfide catalysts under sustainable conditions. *Chem Commun* 51, 7501–7504 (2015).
- The prebiotic source of cysteine is described very late in the manuscript (p. 12/13). This is fine, but I would at least cite the relevant papers much earlier in the main text, e.g. already on p.4.
- Was there a reason to use enantiopure L-cysteine (of course, beyond the clear biological relevance)? Is it necessary? Would using a rac-Cys change anything in terms of the cluster composition, for instance? Any possibly resulting diastereomers could have different properties. If any such changes have been observed, they should be explicitly mentioned, and if not – the choice of L-cysteine appropriately motivated, also in the context of the emergence of chirality (briefly).
- P. 5: “pH 9 to 11, similar to what would be expected in Hadaean alkaline hydrothermal vents” – a reference is needed.
- P.5/6: “whereas solutions with L-cysteine and FeCl₃ concentration held at 5 and 1 mM, respectively, but <0.4 mM Na₂S did not (Fig. 3b and 3c).” – something seems to be missing in this fragment. Also, is <0.4 mM correct?
- P. 6: “clusters did not form when ferric was substituted for ferrous iron” – a bit more discussion is needed concerning what the reason might be
- P. 6: “Low concentrations of ferric iron are congruent with a Hadaean alkaline vent environment” – a concentration range would be useful here
- P. 8 (top) – it is not immediately clear from the way it is described whether the increased cluster stability in the presence of bicarbonate is due to the pH effect or due to the bicarbonate anion itself.
- It would have been fantastic if the authors managed to test the redox properties of the obtained clusters on a model reaction and compared this to ligand-free [4Fe4S] species – even a non-prebiotic model reaction would already be very informative!
- P. 11 (top): what universal metabolic precursors are needed to be either explained or rephrased. Do the cited references (ref. 2 and 12) indeed show the production of oxaloacetate, succinate and alpha-ketoglutarate straight from CO₂? This looks like an overinterpretation.
- P. 11: “UV light is inconsistent with life as we know it” – this is a shortcut, needs rephrasing
- P. 14: the discussion of the applicability of iron-sulfur clusters in the context of protocells and CO₂ reduction is very tricky here. Firstly, is there a reason to invoke CO₂ reduction (therefore C1 compounds) as opposed to CO₂ fixation into higher C products? Secondly, for protocells to arise, higher carbon products are needed (these can in principle come from non-biological Fischer-Tropsch-type processes but some explanation of this logic is needed such that a general chemist can follow the discussion; otherwise, this chicken-and-egg situation may seem confusing to readers from outside the origin of life field).
- P. 21 (the whole experimental section) – the reaction scale as well as the number of replicates in each experiment should be explicitly stated. It should also be stated if any numerical values reported in the paper are taken from a single run or whether they are an average (in the latter case, error analysis is needed). What happens if Cys is added before bicarbonate?

Spontaneous assembly of redox-active iron-sulfur clusters at low concentrations of cysteine

Sean F. Jordan, Ioannis Ioannou, Hanadi Rammu, Aaron Halpern, Lara K. Bogart, Minkoo Ahn, Rafaela Vasiliadou, John Christodoulou, Amandine Maréchal and Nick Lane

Response to reviewers

Reviewer #1 (remarks to the authors)

The paper by Jordan et al addresses the abiotic assembly iron-sulfur (FeS) clusters in conditions compatible to scenarios of the origin of life at alkaline hydrothermal vents. Starting with L-cysteine and FeCl₃(Fe III) - pH between 9 and 11 - 4Fe-4S, 2Fe-2S and Fe are formed. Moreover the authors characterise the FeS centers and show that their redox potential is similar to the ones found in extant ferredoxins. Moreover, FeS were also found to be assembled in the absence of Cysteine, in mixtures containing HCO₃, Fe(III) and Na₂S.

This is a very exciting paper showing the formation of a cofactor essential to all life forms in prebiotic conditions and I am impatiently waiting for the future studies regarding the EPR characterization of the FeS clusters.

We thank R1 for these kind comments and their appreciation of the importance of the work. We agree that EPR characterisation of the FeS clusters will be a worthwhile complementary approach and we hope this can be performed in future either by ourselves or other interested groups using the cluster syntheses described in this paper.

In the condition tested, and in the absence of additional information, Fe(III) from FeCl₃ is in the soluble state. What was the initial redox potential of the mixture (or before adding S-compounds) that allowed the non-precipitation of iron?

As noted in the Methods section, we used ferric chloride hexahydrate (FeCl₃·6H₂O), which was soluble at the concentrations (micromolar to low millimolar range) that we used. We have added a new section on the likelihood of hydrated ferric chlorides being available under alkaline hydrothermal conditions in the Discussion on page 17, lines 395-407. We did not include redox-potential data for individual ions (except

Prof Nick Lane
Professor of Evolutionary Biochemistry
Department of Genetics, Evolution and Environment
University College London
Room 610 Darwin Building
Gower Street, London WC1E 6BT

External phone +44 (0) 20 7679 1385
Internal phone 31385
nick.lane@ucl.ac.uk
www.nick-lane.net

for cysteine alone, shown as before in Fig. 8d), but this is readily available information (for the $\text{Fe}^{2+}/\text{Fe}^{3+}$ couple the midpoint redox potential E° at pH 7 is about +100 mV and for the $\text{S}^{2-}/\text{H}_2\text{S}$ couple it is about +170 mV at pH 7).

Also, please extend the discussion regarding the presence and/or evidences (if available), for the formation of iron-III minerals or the presence of Fe-III in solution (harder I guess) under prebiotic conditions.

We have done so on page 16, lines 395-407.

Again, we thank R1 for their very positive and helpful comments.

Reviewer #2 (remarks to the authors)

In the paper by Jordan et al., the formation of iron-sulphur clusters using the amino acid cysteine is investigated. The main question addressed is the concentration dependences (including the concentrations of Cysteine, Fe, HS-, and pH) required to form these clusters. The techniques used to probe this are uv/vis spectroscopy, Mössbauer spectroscopy, NMR. Cyclic voltametry was used to investigate the reduction potential of the species formed.

From a purely chemical standpoint, the work conducted is important because it sheds light on what a free amino acid might be doing in the cell (though metal ion concentrations there are very low), what a free amino acid might be doing in the environment, and as was the focus of the authors - what free amino acids might be doing before the origin of life, en route to potentially facilitating the origin of life. FeS clusters in biology function in within proteins in catalytic, structural, and redox activities. These are very broad functions, and it seems difficult to imagine biology as we know it without FeS clusters. It is a key question in the origin of life to understand redox transformations, because fundamentally today, biology is an electronic property, with all organisms moving electrons between molecules to do work (i.e. synthesize the polymers that are needed for cell doubling).

Acknowledging the above, significant work has been done towards understanding the abiotic formation of FeS clusters, perhaps most recently experimentally by the Mansy group. One key weakness of this former work – as the authors duly note - is the high concentrations of thiol used in the experiments which is often around 100mM. It seems quite high for “prebiotic” chemistry... Herein lies the value of the authors current approach; by systematically investigating the concentration requirements for FeS cluster formation, the authors are paving new ground and supplying relevant chemical information for the origin of life, when the authors supply evidence that ~800 μM Cys is needed to have a uv/vis observable cluster.

We thank R2 for their appreciation of the importance of this work and its potential impact for the origin of life field. We agree that in a prebiotic context the low concentrations of cysteine that were required for FeS cluster formation are of particular significance and we appreciate that R2 has pointed this out.

Despite being enthusiastic about the provenance of this work, I have major concerns about the conclusions drawn and the data acquisition performed to reach these conclusions which should be

significant enough to prevent publication of the work. I hope my comments below will allow the authors to dive deeper into their question and gain the insight they are working towards. Although the concentration dependences of FeS cluster formation with Cysteine are a valuable part of the paper, the main conclusion of the article is that cysteine affords “monomeric Cys-FeS clusters have structures and reduction potentials exactly equivalent to those found in modern ferredoxins.”. As I outline below, insufficient data are presented to justify this claim. (it is a quibble, but worthwhile to note that it is not clear what “monomeric” means here, since 4Cys are needed for 1 cluster and clusters made of both less and more than 4Fe would be considered to be monomers from a structural perspective)

The crux of our argument is that cysteine monomers in a prebiotic environment could have formed functional FeS clusters prior to the existence of peptides. Although four cysteine molecules are required to form one of the clusters, these cysteine molecules would have existed in a prebiotic ‘monomer world’ scenario as discussed in the text. We have clarified this in the Discussion on pages 14 and 15 to ensure there is no confusion.

Major concerns:

Cysteine: There is a strong theoretical basis against this conclusion. As an example, in a series of papers, the Holm group used the acetylated, methyl amine carboxy substituted version of cysteine (acetyl-cysteine-N-methylamide) to investigate FeS cluster formation from free amino acids, for example: Hill et al, Journal of the American Chemical Society / 99:8 / April 13, 1977 In these works, they specifically used this “capped” version of the amino acid, because free amino acids with their negatively charged carboxyl group and their positively charged amino group are expected to behave fundamentally different from an amino acid that is incorporated into a peptide. Following the work by the Holm group has detailed the critical role that intrinsic/extrinsic solvent effects can have on FeS cluster redox, it is difficult to accept that free Cys would behave like the peptide, or like the “capped” version of Cysteine referenced above. While I have not been able to find the relevant redox properties and cluster formation when free Cys was used, but it is likely published (note that the paper referenced above is part 15 in a series). The authors need to provide a much more detailed background of the literature to convince this reviewer and other readers that Cysteine is a faithful mimic of cysteine containing peptides.

We are very familiar with the extensive work of the Holm group in this area and indeed their work was of particular importance for us when beginning these experiments; especially whether or not they had tried to form FeS clusters using cysteine alone. We did not and still cannot find any evidence they did this work. Their work, and the work of others in the field, is understandably also not carried out under prebiotic conditions, as discussed in the paper. Therefore it does not subtract from the novelty of our paper. We agree that more discussion of this literature would add further clarification to our approach and so we have added to the text in this context at several places, notably in the Introduction lines 71-77.

uv/vis spectroscopy: The authors rely on uv/vis spectroscopy to prove the formation of FeS clusters in the presence of cysteine. This is fair since it is quite easy to use and provides useful information. It is however very unfortunate that only a limited data range is supplied for only a portion of the spectrum - from 370nm-470nm. This is a severe limitation, because single iron “rubredoxin” like complexes usually appear around 330nm (e.g. DOI: 10.1039/c3dt53157k) and 2Fe complexes often show absorbances at longer wavelengths (e.g. Ugulava et al Biochemistry, Vol. 40, No. 28, 2001). Indeed, there are a host of absorbance features that can be observed for different cluster types (Hagen et al. J. Am. Chem. Soc. 1983, 105, 3905- 3913)

With this in mind, it seems dramatically insufficient to report only 100nm of the spectrum.

We are of course familiar with the additional UV-Vis signals that can be diagnostic for FeS cluster types in solution, though the predominant focus of our UV-Vis experiments was simply to provide evidence of cluster formation. But we accept R2's main point here, and have done a major new analysis of the full UV-Vis spectra for the [4Fe4S] clusters, using the ligand field theory approach developed by Galambas et al¹ to determine the concentrations of [4Fe4S] clusters formed by different concentrations of cysteine. These show a linear relationship with cysteine concentration down to 200 μ M cysteine, which is a full three orders of magnitude lower than reported for the short peptide glutathione by Bonfio et al², and indeed two orders of magnitude lower than earlier work, again with glutathione, from Qi et al³. These are presented in a new Fig. 2, with new accompanying text on pages 5 and 6, and we have provided the full-curve-fitting data for all concentrations of cysteine in the SI.

In regard to other FeS clusters, we reaffirm that Mössbauer spectroscopy was the key diagnostic tool for FeS cluster speciation in our approach, as it is superior to UV-Vis in this manner. As noted by Betinol et al⁴, [4Fe4S] clusters absorb strongly and are capable of masking the presence of other clusters on UV-Vis spectroscopy. Because all our preparations contained mixtures of clusters, we therefore established the concentration-dependence of [4Fe4S] cluster assembly on cysteine concentration, and demonstrated in a new figure that this calculated concentration corresponds closely to absorbance at 420 nm, the main signal used previously, which justifies its use in the other analyses originally presented (and which are visually much clearer than using the full curve-fitting model). We feel that others would see this as a useful technique to employ for preliminary analysis of FeS cluster candidate solutions prior to more advance characterisation (Mössbauer/EPR) in the origin of life field.

Mössbauer spectroscopy:

Mössbauer spectroscopy is a very powerful and appropriate tool for studying iron, and it has been used extensively in the study of FeS proteins. As noted in the Pandelia et al review paper cited by the authors "Typical requirements for biological samples of Fe-containing proteins are volumes of 250 - 400 μ L and 57Fe concentrations of 1–5 mM [12]". In the current paper however, the sample was analyzed in a dried state (which the authors call lyophilized, but I suspect is actually simply dried?).

We understand the reviewer's concerns here and we acknowledge that we should have provided further justification for our approach within the text. First and foremost, we stated that the samples were lyophilized because they were. They were not simply dried. Lyophilisation – sometimes called freeze drying – is used to prepare samples that are delicate and in which it is vital to maintain the integrity of the sample structure⁵. This is not the same as simply letting water evaporate from the sample (drying), but instead utilises a three-stage process to gently remove the water. Lyophilisation uses a low

¹ Galambas, A. *et al.* Radical S-adenosylmethionine maquette chemistry: Cx3Cx2C peptide coordinated redox active [4Fe-4S] clusters. *J. Biol. Inorg. Chem.* **24**, 793–807 (2019).

² Bonfio, C., et al., (2017) *Nat. Chem.* **9**, 1229-1234

³ Qi, W. *et al.* Glutathione Complexed Fe – S Centers. *J. Am. Chem. Soc.* **134**, 10745–10748 (2012).

⁴ Betinol, I.O., Nader, S., Mansy, S.S. Spectral decomposition of iron-sulfur clusters. *Analytical Biochemistry*, <https://doi.org/10.1016/j.ab.2021.114269>

⁵ Patel, S. M. & Pikal, M. J. Emerging freeze-drying process development and scale-up issues. *AAPS PharmSciTech* **12**, 372–378 (2011)

temperature dehydration process, which first involves freezing the sample, followed by lowering the pressure and finally removing the ice by sublimation. In this way we are able to preserve the integrity of the delicate sample structure without causing any defects or sample artefacts.

X-ray absorption spectroscopy has shown that the lyophilisation procedure is also fully reversible (Noth et al (2015)⁶ and references therein). Indeed, the authors of this paper use a combination of XANES and EXAFS spectra to show a similar – and crucially, intact, iron coordination environment for the [4Fe4S] subsite of the H-cluster in the freeze dried and in anoxic solution.

Analyzing a dry sample is fundamentally different from analyzing a wet sample. Specifically, there is no reason at all to suspect that the iron coordination in water would represent anything at all of what is occurring in the solution phase. In the analysis of FeS clusters by groups who use Mössbauer, it is to this reviewer's knowledge always done in the wet phase, and this reviewer has never seen a paper where a dried state was used, because it is widely accepted that dry samples are not useful for understanding liquid. Therefore, the analysis of dry samples is inappropriate and cannot be used to analyze the coordination, or redox state of the aqueous clusters.

Mössbauer spectroscopy is the recoilless absorption of a γ -photon by a ⁵⁷Fe nuclei and therefore it is absolutely necessary to fix the atomic nuclei in position so that there is no loss of energy in either the absorption or re-emission of the photon.^{7,8} Any Mössbauer spectroscopy measurements done in the wet phase are actually done on frozen samples, and therefore sub room temperature, and not on wet samples as this would not provide a recoil free absorption. As our measurements have been performed at room temperature, it was not possible to freeze our samples.

Lyophilisation is a very well-established method of preparing FeS clusters for Mössbauer spectroscopy and has been extensively used in a number of published works. For example one of our authors (Lara Bogart) was part of a team that successfully demonstrated structural equivalence between lyophilised clusters of iron oxide and particles measured by room temperature Mössbauer spectroscopy with x-ray diffraction of the clusters growing in solution, highlighting how common and robust is this preparation method⁹.

Furthermore, the analysis of dried samples which have been prepared using other drying methods is also commonplace. Two examples from the origin of life field include the work from the Mansy group as mentioned previously by R2 whereby samples were dried under N₂ prior to analysis¹⁰, and the work by Wenbi Qi et al where samples were dried using a speed vacuum prior to analysis¹¹.

⁶ Noth, J. et al (2015) Lyophilization protects [FeFe]-hydrogenases against O₂-induced H-cluster degradation. *Sci. Rep.* **5** 13978 DOI: 10.1038/srep13978

⁷ Greenwood N N and Gibb T C 1971 Mössbauer Spectroscopy (London: Chapman and Hall) and

⁸ <https://www.rsc.org/membership-and-community/connect-with-others/through-interests/interest-groups/mossbauer/>

⁹ Alec P LaGrow, Maximilian O Besenhard, Aden Hodzic, Andreas Sergides, Lara K Bogart, Asterios Gavriilidis, Nguyen Thi Kim Thanh "Unravelling the growth mechanism of the co-precipitation of iron oxide nanoparticles with the aid of synchrotron X-Ray diffraction in solution" *Nanoscale*, 2019, **11**, 6620-6628

¹⁰ Bonfio, C., et al., (2017) *Nat. Chem.* **9**, 1229-1234

¹¹ Qi, W. et al. Glutathione Complexed Fe–S Centers. *J. Am. Chem. Soc.* **134**, 10745–10748 (2012).

We have stressed this with additional text and references in the revised manuscript, in the Methods as well as the Results on pages 8 and 9.

It might be worthwhile to compare the dried material with Cys to material without Cys, but at any rate, information about Cys-Fe in the aqueous phase is not accessible when dry. It is also improbable that the full reduced 2Fe cluster (blue line) exists in the experiments, where at least partially oxidized (+1, +2) species might be expected.

We again stress that we used a robust and well-established method of sample preparation, which was chosen to be gentle enough to preserve the structure of the clusters in the aqueous phase; moreover, our samples were sealed to prevent any oxidation of the samples on their removal from the glove box.

All spectra were least squares fit to simultaneously fit all sub-spectra within a given sample, with all parameters left able to 'float'. These particular sub-spectra, with close isomer shifts of 0.58 mm/s and 0.60 mm/s, with high quadrupole splitting of 3.33 mm/s and 3.02 mm/s, respectively, are characteristic of high spin ($S=2$) ferrous ions¹². The isomer shifts are perhaps lower than might be expected, which is likely due to the very large electron delocalisation onto the sulphur ligands¹³. However, as we discuss later, this has been observed in the literature for other Fe-Cys [2Fe-2S] bonded clusters. The very slight differences between the two sub-spectra is likely a consequence of minor structural asymmetries around the two valence-delocalised Fe²⁺ pairs, again also reported in the literature¹⁴. The high quadrupole splitting is itself indicative of valence localised Fe²⁺ centres, most probably with quasi-tetrahedral sulphur co-ordination, i.e. mononuclear clusters (Netz et al., 2016). Taken together, these characteristics are all reminiscent of the high-spin cluster [2Fe-2S]⁰ – also known as the Rieske protein^{15,16}). This is what we – and others – have observed, and so we do not agree that our observation is improbable.

We believe that the structure of FeS clusters and complexes is independent of whether the cluster is in the aqueous phase or not, provided that the route taken to remove the water is compatible with the preservation of the structure. Indeed, many of the literature parameters of rubredoxins and ferredoxins have been arrived at by measuring naturally occurring clusters found in spinach, for example, in K. K. Rao, R. Cammack, D. O. Hall and C. E. Johnson "Mossbauer Effect in Scenedesmus and Spinach Ferredoxins" *Biochem J.* 1971 Apr; 122(3): 257–265.

Quickly checking the 2Fe2S clusters in the Pandelia paper, I find that the velocities associated with the absorbances are not at the values shown in the paper for the 2Fe species. Going further, both $|\Delta EQ|$ (mm/s) and δ (mm/s) values are different from what Pandelia et al reported in their table 1 in

¹² Gütlich, P. (1975) in *Mössbauer Spectroscopy*, ed. Gonser, U. (Springer, Berlin), Vol. 5, pp 53–96

¹³ Johnson, C.E., Applications of Mossbauer Effect in Biophysics. *Journal of Applied Physics*, 1971. 42(4): p. 1325-&.

¹⁴ Netz, D.J.A., et al., *The conserved protein Dre2 uses essential [2Fe-2S] and [4Fe-4S] clusters for its function in cytosolic iron-sulfur protein assembly.* *Biochemical Journal*, 2016. **473**: p. 2073-2085.

¹⁵ Fee, J.A., et al., Purification and Characterization of the Rieske Iron-Sulfur Protein from *Thermus-Thermophilus* - Evidence for a [2Fe-2S] Cluster Having Non-Cysteine Ligands. *Journal of Biological Chemistry*, 1984. 259(1): p. 124-133.

¹⁶ Leggate, E.J., et al., *Formation and characterization of an all-ferrous Rieske cluster and stabilization of the [2Fe-2S](0) core by protonation.* *Proceedings of the National Academy of Sciences of the United States of America*, 2004. **101**(30): p. 10913-10918.

comparison to table 1 in the current paper. This is a major inconsistency and makes the data and interpretation suspect, even in the absence of the major problems associated with analyzing and interpreting coordination in dry samples. Perhaps the numbers simply do not match because they are reporting different things.

This is to be expected, as all of the spectra presented within the Pandelia paper are from measurements taken at 4.2K whilst our measurements were performed at room temperature, and so this is not a like for like comparison. The second order Doppler shift (SODS) means that the isomer shift increases as the temperature decreases, and so temperature corrections must be performed in order to compare like with like¹⁷.

Going further, both $|\Delta EQ|$ (mm/s) and δ (mm/s) values are different from what Pandelia et al reported in their table 1 in comparison to table 1 in the current paper. This is a major inconsistency and makes the data and interpretation suspect, even in the absence of the major problems associated with analyzing and interpreting coordination in dry samples. Perhaps the numbers simply do not match because they are reporting different things.

The values reported in table 1 of Pandelia et al., are taken from samples measured at 4.2 K and this is therefore not a like for like comparison. Such differences in reported values are a direct consequence of the effect of temperature (as discussed in our previous answer), and have not arisen due to the differences in sample preparation. Our values agree very well with other reported data on Fe²⁺-Cys, reported on by Leggate et al (PNAS, 2004) and references therein, for example in Table 1 [REDACTED]

¹⁷ Jeppe Fock, · Lara Katrina Bogart, · Oliver Posth, · Mikkel Fougth Hansen, · Quentin A. Pankhurst, and · Cathrine Frandsen “*Uncertainty budget for determinations of mean isomer shift from Mössbauer spectra*”, *Hyperfine Interact* (2016) 237: 23 DOI 10.1007/s10751-016-1253-1

Note, these data were measured at 160 K. Our data are also consistent with a number of other articles referenced within Leggate et al., including:

- tetrahedral Fe^{II}S₄ sites in the [2Fe-2S]⁰ cluster in Aquifex aeolicus ferredoxin ($\delta = 0.71 \text{ mm}\cdot\text{s}^{-1}$, $\Delta E_Q = 2.75 \text{ mm}\cdot\text{s}^{-1}$ at 4.2 K) ¹⁸
- rubredoxin ($\delta = 0.7 \pm 0.02 \text{ mm}\cdot\text{s}^{-1}$ at 4.2 K) (Fe_H has $\delta = 0.78 \text{ mm}\cdot\text{s}^{-1}$ and $\Delta E_Q = 2.24 \text{ mm}\cdot\text{s}^{-1}$ (160 K) ¹⁹
- ferrous ion in the [2Fe-2S]¹⁺ clusters of BtRp ($\delta = 0.69 \text{ mm}\cdot\text{s}^{-1}$, $\Delta E_Q = 2.87 \text{ mm}\cdot\text{s}^{-1}$, 160 K) and TtRp ($\delta = 0.65 \text{ mm}\cdot\text{s}^{-1}$, $\Delta E_Q = 2.81 \text{ mm}\cdot\text{s}^{-1}$, 230 K) (Fee et al, 1984).

At this point we feel it is important to note that whilst there is an incredible range of literature regarding Mössbauer spectroscopy studies into Fe-S clusters in the form of a range of natural source rubredoxins and ferredoxins as well as laboratory synthesised materials, assignment of sub-spectra is made significantly more difficult as numerous papers reference Mössbauer parameters taken from spectra obtained at low temperatures, and often seemingly without any second-order doppler shift (SODS) correction applied to the value of the isomer shift. Furthermore, not all spectra are folded relative to a-Fe which further complicates the interpretation.

Therefore, whilst the literature is rich in data, there is a lack of consistent studies on spectra collected at room temperature in zero applied field. With this in mind, the interpretation presented here is based upon a number of scientific papers in which the Mössbauer spectroscopy was either SODS corrected (or

¹⁸ Yoo, S. J., Meyer, J. & Münck, E. (1999) *J. Am. Chem. Soc.* 121, 10450–10451

¹⁹ Schulz, C. & Debrunner, P. G. (1976) *J. Phys. Colloque* 12, 153–158.

from which we have applied a rudimentary SODS correction using Figure 4 (Fock et al., 2016, REDACTED) and in which the deduced Fe-S structure was confirmed using a range of complementary techniques.

cyclic voltametry:

Cyclic voltametry is a powerful tool for understanding the tendency of FeS clusters (and any number of other species) to accept and give electrons. The Hill paper cited above used the technique to good effect, for example. In the current work, the authors took samples from the glove box and “rapidly transferred to the potentiostat “. This seems rather unfortunate, since it is almost certain that oxygen is now a major factor in the observations.

We had perhaps not made it sufficiently clear that we had first purged with N₂ for 15 minutes before transfer to the potentiostat, so we do not believe this was an issue and have clarified that in the revised MS (page 29, lines 731-33).

Putting aside the concern of O₂ however are the interpretations. The authors say that “edge” features exist which show that the clusters have potentials comparable to modern ferredoxins (~600mV vs AgCl written by the authors), but I cannot find the “edge” features, and cannot conclude with the authors on this statement. Instead, I suspect that the authors are seeing ferrous/ferric transitions, together with multiple redox active sulfur steps, especially in the more oxidized regions of the cathodic scans. It's very worthwhile here to visit the FeS literature, for example: Bura-Nakic' et al. Electroanalysis 2011, 23, No. 6, 1376 – 1382.

We accept R2's comments here and have revised this section accordingly, pointing out the need for more work on individual complexes, which will be done in a future paper. We now make it clear that the cathodic potential shifts to more negative values (from -310 to -400 mV vs Ag/AgCl electrode) when a small proportion (9-18 %) of [4Fe4S] clusters are present. This revised analysis on page 15 does not call on any debatable 'edge' features but instead explains why substantive further work is needed. However we feel that this is a separate and large-scale study in itself, which we will undertake in the next year.

minor comments:

with regards to citations 1 and 9 cited for claiming that ferredoxins are ancient. It is worthwhile to note that (1) conflated spatial proximity within proteins for antiquity - an interesting yet unproven hypothesis and that (9) seems to have been proven largely incorrect (doi:10.1093/molbev/msaa089). The authors might be better off simply claiming that it's hard to imagine biology without these clusters, as I have done in the beginning of this review.

We now add in a little more detail here in the Introduction and cite the paper by Berkemer and McGlynn contesting the phylogenetic findings of Weiss et al.

“For any form of selection to operate, there had to be a continuous succession of steps leading from Fe²⁺/Fe³⁺ and S²⁻ mixing in solution (which spontaneously crystallize to form FeS minerals) to early cells, where discrete FeS clusters were assembled within proteins instead.” This sentence is very unclear to me, can the authors work on it to make it more specific and concrete?

We have simplified and clarified this passage in the revised manuscript.

“Moreover, the headspace above the sample within the NMR tube is of a larger volume than the sample itself, so it would have taken significantly longer for O₂ to diffuse into the sample than in the UV-Vis cuvette.” This is another unclear sentence, since volume might not be related to the rate of oxidation...

This was indeed unclear. The suggestion here is that the large anoxic headspace above the sample itself is sufficient to delay O₂ penetration on exposure to aerobic conditions when compared to the relatively small headspace of the UV-Vis cuvettes. The dimensions of the NMR tube may also contribute to this delay as the narrow dimensions of the tube may also delay O₂ penetration while the surface area of the sample solution that is in contact with the headspace above is also reduced when compared with the cuvette. We have expanded on this in the revised manuscript (lines 263-64).

“Both 3.5 and 5 mM solutions provide voltamograms with ΔE_p values of 55 and 63 mV respectively, which correspond to a one-electron transfer” No, this suggests that the charge transfer has an n value of 1 in the Nernst equation. It means that the number of protons equal the number of electrons:

When Ox+ne⁻ +mH⁺→Red the slope at 25 °C will be: - (m/n) Å~ 59 mV/pH.

consider the case of no protons being involved in the reaction; the slope would be zero.

This is correct and we have revised the MS to specify a Nernst n value of 1. We had previously written this loosely to be more understandable to a wider audience unversed in CV, based on Sandford et al. (2019)²⁰ who write:

²⁰ Sandford, C., et al., (2019), Chem. Sci. 10, 6404

“The separation of the two peak potentials for a reversible redox event with fast electron-transfer kinetics is equal to $\sim 60/n$ mV at room temperature, where n is the number of electrons transferred in the redox event... Accordingly, the variation in peak separation upon changing the CV scan rate can be used to quantify the heterogeneous electron transfer rate constant between the electrode and redox-active species”

“orders of magnitude lower than that of the polymers used in earlier work¹⁷” It is correct that Mansy’s group uses high concentrations, and as I pointed out above, a major strength of this current work is that it sets out to explore the concentration limit for FeS cluster formation, which to the knowledge of the current reviewer has not been done.

I quickly checked work from Cowan’s group “[dx.doi.org/10.1021/ja302186j](https://doi.org/10.1021/ja302186j)” which seems to be a forerunner of Mansy’s work and found 10mM. But in a recent paper that discussed FeS clusters I also found that FeS clusters seemed to have been made at “750 mM Ferredoxin peptide”, which is equal to the concentration of cysteine used in the current work (though it is very important to remember that the ferredoxin forms multiple linkages with the cluster from a single molecule, and cysteine probably is not formally a ligand but instead requires 4 per cluster (if excluding the possibility of NH_3^+ and carboxyl ligation, which could be important). It is worthwhile to ask if anyone has done a concentration dependence test for peptides as well...

We thank the reviewer for their appreciation of the importance of our investigation of cysteine concentration limits for FeS cluster formation, which we believe is imperative in this prebiotic context. The 10 mM Glutathione used in the Cowan paper is low although it is still a peptide, whose provenance remains unclear in an origin of life context, as is the ferredoxin peptide regardless of whether the cysteine concentration is equivalent to what we have presented – which in fact, with the additional analyses suggested by R2, we have now lowered to 200 μM cysteine. Of course the Fd peptide is also presumably of biological origin (no reference provided) and therefore is again problematic from a prebiotic perspective. As all reviewers here have agreed, our work is novel and important primarily for the fact that we employ very low concentrations of a prebiotically plausible, biologically relevant, monomeric amino acid to produce FeS clusters that could have functioned as precursors to enzymes at the earliest stages of the origin of life. We have referred to the Cowan paper in the revised version of this manuscript and included it as part of our discussion on the limitations of previous work in the field.

Methods:

problems with oxygen: in the uv/vis: sealing with parafilm is not sufficient to minimize oxygen reaction. It means that all of the results obtained with uv/vis have a strong oxygen component, which is in turn important for which clusters would be formed. Measurements in cells with screw caps, or better yet, within the anaerobic chamber, would be preferable.

We agree that parafilm is not sufficient to completely prevent oxygen penetration, despite purging with N_2 ; however the purpose here was to minimise interference prior to analysis. We would of course have ideally analysed these samples in the anaerobic chamber. Unfortunately we did not have access to an instrument that could be installed and operated within the anaerobic chamber. This is another reason why we took limited diagnostic information from UV-Vis analyses, using it predominantly as a tool for identifying candidate solutions to be analysed by Mössbauer spectroscopy where we could be confident in our interpretation of the results providing more detailed data on FeS cluster speciation as discussed above.

in the potentiostat: it again seems that the measurements are going to be dominated by the presence of O₂.

As noted above, we took precautions to minimize exposure to oxygen and have clarified the text to highlight the purging with N₂. Clearly the FeS clusters were stable for hours outside the anaerobic hood (see Fig. 6), rather days inside the hood (SI. Fig. 13), but we reiterate that our analyses were done immediately upon transferring from the hood, having been purged with N₂ as effectively as possible. The fact that we detect clear structures for [4Fe4S] clusters by both full curve-fitting on UV-Vis spectra and Mössbauer spectroscopy supports our belief that the measurements are not dominated by the presence of O₂.

Figure 1: This reviewer can't think of a ferredoxin that has both a 4Fe and a single Fe unit. Can the authors please indicate the pdb used in generating this?

We have added this information to the figure legend in the revised manuscript.

We sincerely thank R2 for their extremely helpful and detailed analysis and for pushing us to do more thorough analyses of the full UV-Vis spectra, which we agree substantially improve the paper. We have added many detailed clarifications to the paper which again improve it immeasurably. Our sincere thanks for taking the time and effort to help us improve the paper. We hope they will agree that the substantially revised MS is worthy of publication in *Nature Communications*.

Reviewer #3 (remarks to the authors)

Iron-sulfur clusters are considered as central cofactors in biochemistry, and also ones whose evolutionary history is likely to date back to the very emergence of life from prebiotic chemistry. It is unknown, however, how Fe-S clusters may have been produced under prebiotic conditions in a way robust and simple enough to have enabled their facile incorporation into the earliest forms of nascent biochemistry. The most prominent example to date is the one reported in Nat Chem by Bonfio et al, and involves UV light-mediated photooxidation of Fe²⁺ driving the synthesis of [2Fe-2S] and [4Fe-4S] clusters, stabilised by cysteine containing tripeptides such as glutathione.

In the present paper by Jordan et al, the synthesis of [2Fe-2S], [4Fe-4S], as well as mononuclear clusters is shown under much simpler conditions (no UV light needed), at low Fe²⁺ and Cys concentrations and at a mildly alkaline pH – which is all in line with the hydrothermal vent hypothesis explored by the authors of the manuscript. The iron-sulfur clusters are characterised by Mössbauer spectroscopy, UV-Vis spectroscopy and ¹H NMR, and their redox properties are examined using cyclic voltammetry.

The analytical part of the study appears very clear and thorough, and the results well-documented and easy to follow. Only for this reason, i.e. the demonstration that iron-sulfur clusters can be formed using such remarkably simple chemistry, I am of the opinion that this paper deserves publication. Without the need to invoke the UV light (therefore a process that could only have taken part in very shallow waters) or the presence of complex thiols like glutathione, this study indicates that the simplest forms of ferredoxin-type “protocofactors” may have predated the emergence of stable oligopeptides or robust

lipid membranes – and why not already taken part in catalysis? Here, I need to stress that this is not to say the two approaches are mutually exclusive, even if one is likely to have preceded the other, and I think it would be worthwhile to make it more explicit in the text.

We thank R3 for the very positive comments on our work and their support of its publication in *Nature Communications*. We are pleased that R3 finds the analytical approach to be of high quality and that they grasp the importance of the work. R3 also makes an interesting point with regards to the possibility of prebiotic FeS cluster catalysis both with and without the presence of oligopeptides or membranes. We have clarified this point in the Discussion on page 14 (lines 336-37)

Before publication, I suggest that the authors address the following questions and remarks that should hopefully improve the manuscript. I will list them below in a chronological order:

- Page 3 – the following paper by de Leeuw and co-workers could be added to the discussion on the use of greigite in CO₂ reduction and fixation: Roldan, A. et al. Bio-inspired CO₂ conversion by iron sulfide catalysts under sustainable conditions. Chem Commun 51, 7501–7504 (2015).

We agree that this is a helpful addition here and have included it in the revised manuscript.

- The prebiotic source of cysteine is described very late in the manuscript (p. 12/13). This is fine, but I would at least cite the relevant papers much earlier in the main text, e.g. already on p.4.

We tried to fit this into the introduction as suggested but it disturbed the structure of our argument, so we have referred the reader to the discussion (line 82) to make it clear at that point that we address the availability of cysteine later on.

- Was there a reason to use enantiopure L-cysteine (of course, beyond the clear biological relevance)? Is it necessary? Would using a rac-Cys change anything in terms of the cluster composition, for instance? Any possibly resulting diastereomers could have different properties. If any such changes have been observed, they should be explicitly mentioned, and if not – the choice of L cysteine appropriately motivated, also in the context of the emergence of chirality (briefly).

We did not investigate both enantiomers of cysteine in this work. We used 99% L-cysteine only (which likely contains a negligible amount of D-cysteine). This decision was based on the biological relevance as the reviewer has correctly noted. We feel that chirality is a separate question to what we have investigated in this current work. In any case it is unlikely that the presence of D-cysteine would have any effect on FeS cluster formation. The chemical properties (e.g. charge, polarity etc.) that could potentially have an effect on coordination of these simple FeS clusters are identical for both L- and D-cysteine. However, it is conceivable that the chirality may impact the interaction of FeS clusters with more complex structures such as membranes or indeed proteins in later stages of prebiotic chemistry. To this end, it would be worthwhile to investigate mixtures of enantiomers in future work addressing these issues. We thank the reviewer for this observation.

We have added a note on chirality in page 16 lines 392-94.

- P. 5: *“pH 9 to 11, similar to what would be expected in Hadaean alkaline hydrothermal vents” – a reference is needed.*

Added to the revised manuscript.

- P.5/6: *“whereas solutions with L-cysteine and FeCl₃ concentration held at 5 and 1 mM, respectively, but <0.4 mM Na₂S did not (Fig. 3b and 3c).” – something seems to be missing in this fragment. Also, is <0.4 mM correct?*

<0.4 mM Na₂S is correct. The data shows that below this concentration of Na₂S FeS clusters will not form, even in the presence of sufficient concentrations of L-cysteine and FeCl₃. We reworded this statement in the text to clarify our interpretation.

- P. 6: *“clusters did not form when ferric was substituted for ferrous iron” – a bit more discussion is needed concerning what the reason might be*

We have extended this discussion in the revised manuscript (lines 144-49).

- P. 6: *“Low concentrations of ferric iron are congruent with a Hadaean alkaline vent environment” – a concentration range would be useful here*

We have discussed this in a new paragraph on page 17.

- P. 8 (top) – *it is not immediately clear from the way it is described whether the increased cluster stability in the presence of bicarbonate is due to the pH effect or due to the bicarbonate anion itself.*

We interpret this increased stability as being due to the bicarbonate ion itself and not the pH effect. We have reworded this text in the revised manuscript to provide clarity.

- *It would have been fantastic if the authors managed to test the redox properties of the obtained clusters on a model reaction and compared this to ligand-free [4Fe4S] species – even a non-prebiotic model reaction would already be very informative!*

We agree that this would be a worthwhile experiment to perform, and we plan to do this as part of future work. However we feel that the amount of experimental work we have performed is already significant and was in fact performed right up until the closure of laboratories here in the UK due to the Covid-19 pandemic. Unfortunately it was not possible to perform further work of this nature after this, as I am sure the reviewer can appreciate. To do this work properly (which we intend to do) would be a full paper in itself. That would only distract from the key important messages of this paper, which is that the single amino acid cysteine can form biological redox-active FeS clusters in alkaline solution under plausible prebiotic conditions at feasibly low concentrations of the ions and ligands needed.

We feel that the redox potential values obtained are sufficient to allude to the catalytic possibilities of these FeS clusters. We have made it clear that this is an important future direction, but goes beyond the focus of this paper, in the Results on pages 12-13 and the Discussion on page 15-16.

- P. 11 (top): what universal metabolic precursors are needed to be either explained or rephrased. Do the cited references (ref. 2 and 12) indeed show the production of oxaloacetate, succinate and alpha-ketoglutarate straight from CO₂? This looks like an overinterpretation.

The references cited for this statement were incorrect. We should have referenced 8 and 13 which do indeed show that the reduction of CO₂ can produce these key metabolic precursors. However, it is a two-step synthesis as opposed to a 'one-pot' reaction as implied by our text. We have expanded the text on this subject in the revised manuscript so as to make this point clearer and more balanced (lines 325, 376 and 441-2).

- P. 11: "UV light is inconsistent with life as we know it" – this is a shortcut, needs rephrasing

A fair point, which we have addressed in the revised manuscript (lines 338-9).

- P. 14: the discussion of the applicability of iron-sulfur clusters in the context of protocells and CO₂ reduction is very tricky here. Firstly, is there a reason to invoke CO₂ reduction (therefore C1 compounds) as opposed to CO₂ fixation into higher C products? Secondly, for protocells to arise, higher carbon products are needed (these can in principle come from non-biological Fischer-Tropsch-type processes but some explanation of this logic is needed such that a general chemist can follow the discussion; otherwise, this chicken-and-egg situation may seem confusing to readers from outside the origin of life field).

We agree that our text on this point is geared towards an origin-of-life audience and should be explained further to satisfy the broader scientific audience of *Nature Communications*. However we feel that the MS is now about as long as it can be, so we have restricted ourselves to a brief clarification (lines 441-2) to give a perspective on our approach. We hope this is sufficient.

- P. 21 (the whole experimental section) – the reaction scale as well as the number of replicates in each experiment should be explicitly stated. It should also be stated if any numerical values reported in the paper are taken from a single run or whether they are an average (in the latter case, error analysis is needed). What happens if Cys is added before bicarbonate?

We now state the number of replicates (triplicate). Cysteine was never added before bicarbonate as we were attempting to replicate a realistic environmental scenario as best as possible. It is hard to imagine a vent fluid containing cysteine into which bicarbonate is subsequently introduced as compared to the opposite sequence of events. In an ocean vent the fluid would probably be much more likely to contain bicarbonate ions initially (as well as multiple other ionic species).

Once again, we thank R3 for their very positive and constructive review, and for raising a number of important points which have improved the paper considerably.

REVIEWERS' COMMENTS

Reviewer #2 (Remarks to the Author):

please find responses in the attached document.

Reviewer #3 (Remarks to the Author):

The Authors have addressed all of my comments from the first round of review. The manuscript has been considerably improved by the additional discussion added to the main text, as well as the more precise description of the presented data. Even though it was not a direct result of my own review, the additional Mossbauer spectroscopy discussion and data are exceptionally valuable for the full understanding of the present study by the reader.

Taking the above into consideration, the paper is now fit for publication "as is", and I am looking forward to the Authors' subsequent reports on the use of [4Fe4S] clusters in catalysis as well as the insights into their redox properties with and without the small organic ligand.

Reviewer #2 (remarks to the authors)

This is a much improved version of the paper. I still have major concerns about the Mössbauer data, but since the importance of the work is independent of this, I suggest the authors revise as I suggest below to qualify their work. It is important to emphasize that the obtained results are provisional in nature; frozen samples should be analyzed, though that is not critical for this specific work.

We thank R2 for their open-mindedness on this. We have done as they suggest (see detailed responses below). We'd also like to sincerely thank R2 for devoting so much time and expertise to this paper; it is much improved for their input. Thank you!

In addition, the discussion of the redox potentials needs revision. We should not discuss only the cathodic potential or anodic potential. The reason "redox potentials" is used by many authors is that it is shorthand for implying that midpoint potential is being discussed - that is, the potential half way between the cathodic and anodic potentials at reversible equilibrium. Currently, only the cathodic potentials are discussed, and it is misleading to compare these to midpoint potentials of FeS clusters. Please revise accordingly.

Likewise, we have done this as requested (again more details below).

Mössbauer:

I checked 2 of the papers that the authors cite to justify the use of lyophilized samples, but both of them studied frozen "wet" samples. It is difficult to find a literature precedence of thiol-ligated FeS clusters studied after lyophilization:

Leggate et al 2004

"Purified 57Fe-BtRp was concentrated to 1.5–2.0 mM for preparation of the all-ferrous Rieske cluster at 300 – 600 μ M."

And Rao 1971:

From Pandellia 2015 we can read:

"Typical requirements for biological samples of Fe-containing proteins are volumes of 250 - 400 μ L and 57Fe concentrations of 1–5 mM [12]. Although some biological materials with natural abundance Fe

Prof Nick Lane
Professor of Evolutionary Biochemistry
Department of Genetics, Evolution and Environment
University College London
Room 610 Darwin Building
Gower Street, London WC1E 6BT

External phone +44 (0) 20 7679 1385
Internal phone 31385
nick.lane@ucl.ac.uk
www.nick-lane.net

have been studied by Mössbauer spectroscopy, it is generally required that samples be enriched with ^{57}Fe ."

*Münck discussed sample preparation in 1978 article "Mössbauer Spectroscopy of Proteins: Electron Carriers" and wrote that samples must be "solid, i.e., frozen solution, lyophilized materials (biologically not very desirable), or single crystals have to be used." Which gets at the same point that I raised. I remain unconvinced that results obtained from lyophilized samples can be extrapolated to what may exist in the liquid state. The only paper I have seen that suggest this to be the case is the Noth *et al* paper that the authors cite in their rebuttal. I suggest the authors cite this paper also in their main text (the result is mentioned at line 169, but the citation is left out), and qualify their Mossbuaer results in the introduction to this technique by clearly saying that FeS clusters in proteins are typically studied as frozen, hydrated samples, and then justify why it might be (or might not be!) that a dry sample would be different from a wet sample...*

There is a misunderstanding here. We apologize if we were not clear. We did not cite the papers mentioned here by R2 to justify the procedure of lyophilization – earlier in that response we had used several other papers to justify this approach (notably Noth *et al* as noted by R2, which should have been cited in the MS but was eliminated by mistake). We cited the papers that R2 raises here to demonstrate that the structures of FeS clusters determined by our method corresponded closely to those reported in the literature using other methods, notably freezing. In the case of Pandelia *et al*, our point was that the values vary slightly because their analysis was carried out at 4.2K, whereas ours was carried out at room temperature; that difference is expected to give slightly different values. Analyses of samples measured at room temperature gave values close to those we report.

In our view, the fact that Mössbauer analysis verified cluster coordinations after lyophilisation is a very strong indication that the samples were not damaged or altered dramatically during preparation; the only alternative is to assume that the sample preparation itself formed the clusters, which seems at the least improbable. Our interpretation is corroborated by the UV-Vis spectroscopy for [4Fe4S] clusters, as well as the differences in redox potential by CV for various mixes of cluster predicted by the Mössbauer analysis. We added a short section on page 6 that defends our approach briefly, including an additional reference by Lorent *et al*. [*Angew. Chemie Int. Ed.* **60**, 15854–15862 (2021)], who show equivalent activity after lyophilization and reconstitution of O_2 -tolerant [NiFe]-hydrogenases to freshly isolated enzyme, indicating that "metal cofactors and amino acid side-chains responsible for proton/electron transfer were not altered by lyophilization." Nonetheless, we acknowledge that our method is not fully validated, as requested by R2.

Minor comment: Reference 35: Is the reference correct? This paper does not seem to deal with reversible FeS cluster lyophilization.

Whoops. No; this should have been Noth *et al*. Thanks for pointing out. Corrected now.

Reduction potentials:

When discussing the reduction potentials of the produced species that show reversible electron transfer, the formal reduction potential should be estimated using:

If only the anodic or cathodic potential is given, it might be a little misleading for a reader. If I look at figure 8, I estimate panel A and B to correspond to E_0 values of -200 and 0, which would be 0 and + 200 vs SHE.

We have added six lines discussion of these points on page 16. We had not discussed the midpoint potentials before as the anodic potentials are either non-reversible or quasi-reversible, especially at the higher cysteine concentrations. Reversibility is favoured at higher scan rates, so we now use the higher scan rates in Supplementary Figure 14 to estimate the midpoint potential for 3.5 mM cysteine, which is about -250 mV vs Ag/AgCl electrode, or -50 mV vs SHE, close to that estimated by R2.

I do not know of any 4Fe4S clusters in biology that have this positive of a potential, except perhaps for the HiPIP clusters. Therefore, I find it difficult to accept the statement on line 308 that “These Cys-FeS clusters are equivalent to those found in extant proteins, notably ferredoxins”. They are not. Please modify the sentence to reflect this.

We agree that this is too high to reduce CO₂ and state this clearly in the text. We have modified the sentences on the reduction potentials of ferredoxins to mention the HiPIPs. We also slightly expand and reorganize our discussion on composite peaks to make clear why the midpoint reduction potentials for [4Fe4S] are likely to be lower than the composite peaks, and worthwhile exploring in systematic future work. We believe that these changes address R2’s concerns while explaining better why the redox values recorded are still important.

Other comments that need revision:

Line 311: please modify the sentence; ferredoxins do not occur with this high of a potential.

Modified as requested; see comments above.

Line 312: It’s not clear what a composite cathodic signal is, and what the concentration dependence being referred to is.

This had been discussed in the preceding paragraph in the Results section and in more detail later in the Discussion (page 16); this paragraph was intended as a summary of the overall results. However we have clarified what we mean here, as requested, and also clarified the later discussion to make this point as clear as possible.

Line 325: Greigite was not used in the experiments by Muchowska et al (reference 8) and the other paper cited is a review; please revise this sentence accordingly.

This is correct and was an oversight on our part. We have now clarified exactly which carboxylic acids were formed with a greigite catalyst in the experiments of Preiner *et al.* and Roldan *et al.*, and given a slightly wider context for other Krebs cycle intermediates in Muchowska *et al.*

Line 338: “synthesis of Glu-FeS clusters also requires UV light”. It does not generally. Instead, the cited paper demonstrated that ferrous iron could be oxidized, and sulfur liberated from thiols by uv light, which could then act as reagents in FeS cluster formation. Please revise the sentence.

Yes, we agree, and simply deleted this sentence as it did not add to our main point about monomers.

Generally, such high concentrations of a polymer are un-needed, as I stated in my previous review, and is also known from the literature in general, where low concentrations of peptide are routinely used in

reconstitution experiments (for example see Hoppe et al Biochimica et Biophysica Acta 1807 (2011) 1414–1422, which used pretty standard methodology and was less than mM during reconstitution).

We have clarified that we are referring specifically to prebiotically relevant work. The method in Hoppe *et al*, for example, uses a peptide of 33 amino acids, which is not relevant to the early period of prebiotic chemistry that we are discussing; even glutathione, with three amino acids, is of questionable prebiotic provenance. We are perplexed by the ‘less than mM’ quantities mentioned by R2; the concentration of maquette in these experiments was 50 mM, so 2-3 orders of magnitude higher than the lower cysteine concentrations used in our experiments. Moreover the preparation method is complex and certainly not applicable to prebiotic chemistry.

One 350: I’m not sure what “three monomeric cysteine molecules” means here - the cited paper investigated short polymers, and in the case of the 16 mer nesting was observed, though for short periods of time.

The paper also considered monomeric cysteine. See for example the discussion of ‘three free cysteine amino acids on page 517 and Figure 1 which presents “key structural features of aqueous amino acid models: 2Cys A, B), 3Cys C, D), and 3Cys + 4Gly E, F) that are to be used to characterize the peptide models”. In this 50 ns molecular dynamic simulation, no [4Fe4S] clusters were formed. However, we have modified our text slightly as the meaning of ‘three cysteines’ is unclear without more explanation of the molecular dynamic context, which is unnecessary to our point.

Line 356: “These monomer concentrations are two to three orders of magnitude lower than that of the polymers used in earlier work^{23,62}” Yes, but it is unfair to compare only to the cited references. As I have noted in this review and also the previous review, it is routing to form clusters under reconstituting conditions in the 100’s of micro molar range. It is actually preferred to work at these low concentrations, because iron and sulfide are not soluble together above ~200µM. Please revise this sentence or simply remove it. It is simply not sure that this current work has demonstrated FeS cluster formation at orders of magnitude lower concentration than that in polymer studies; the work from the cited papers perhaps being exceptions.

See comments above. We have clarified to refer specifically to prebiotically relevant work. We have also made it clear that it is the concentration of the ligand (cysteine or glutathione) that we are specifically referring to rather than the iron and the sulfide. As an aside, we would expect specifically engineered larger maquettes (of 30+ amino acids) to form [4Fe4S] clusters at lower concentrations, simply because the maquette provides a nest for [4Fe4S] cluster formation. Even glutathione would be expected to do this to some extent. So the fact that cysteine readily forms [4Fe4S] clusters at low concentrations, while tending to form [2Fe2S] or monomeric clusters at higher concentrations, is a point worth making.

line 356. revise this discussion of reduction potentials - we are not only talking about Ec, but E0 when we talk about mid point potentials. Simply put, although the finding of FeS clusters with cysteine is exciting, the reduction potential of this is insufficient to operate in low potential arms of metabolism or prebiotic chemistry.

Done. But see also points raised above about composite peaks with multiple species.

Line 401: The oceans still would have been ferruginous, but the dissolved iron concentration would have been lower. The in situ oxidation may have happened after the GOE, so it's not clear how this "provides a more compelling source of ferric iron"

We have modified this passage. The point is not that ferric iron was formed in BIFs, but rather that modelling shows that alkaline hydrothermal conditions can in principle oxidise ferrous to ferric iron within the vent system itself; from this point of view it doesn't matter whether the ferrous iron derives from the oceans or the vent system itself.

Once again we thank R2 for their extremely helpful and constructive comments that have improved the paper considerably.

Reviewer #3 (remarks to the authors)

The Authors have addressed all of my comments from the first round of review. The manuscript has been considerably improved by the additional discussion added to the main text, as well as the more precise description of the presented data. Even though it was not a direct result of my own review, the additional Mossbauer spectroscopy discussion and data are exceptionally valuable for the full understanding of the present study by the reader.

Taking the above into consideration, the paper is now fit for publication "as is", and I am looking forward to the Authors' subsequent reports on the use of [4Fe4S] clusters in catalysis as well as the insights into their redox properties with and without the small organic ligand.

We sincerely thank R3 for their support, and their time and attention in reviewing the MS.